# A genome engineering resource to uncover principles of cellular organization and tissue architecture by lipid signaling

Deepti Trivedi, Vinitha CM, Karishma Bisht, Vishnu Janardan, Awadhesh Pandit, Bishal Basak, Shwetha H, Navyashree Ramesh, Padinjat Raghu*

National Centre for Biological Sciences-TIFR, GKVK Campus, Bangalore, India

**Abstract** Phosphoinositides (PI) are key regulators of cellular organization in eukaryotes and genes that tune PI signaling are implicated in human disease mechanisms. Biochemical analyses and studies in cultured cells have identified a large number of proteins that can mediate PI signaling. However, the role of such proteins in regulating cellular processes *in vivo* and development in metazoans remains to be understood. Here, we describe a set of CRISPR-based genome engineering tools that allow the manipulation of each of these proteins with spatial and temporal control during metazoan development. We demonstrate the use of these reagents to deplete a set of 103 proteins individually in the *Drosophila* eye and identify several new molecules that control eye development. Our work demonstrates the power of this resource in uncovering the molecular basis of tissue homeostasis during normal development and in human disease biology.

*For correspondence:
praghu@ncbs.res.in

**Competing interests:** The authors declare that no competing interests exist.

## Introduction

Phosphoinositides (PI) are low-abundance phospholipids implicated in the regulation of many important biological processes including cell polarity, cell migration, aging, growth, and development (*Balla, 2013*). The head group of the parent lipid phosphatidylinositol can be phosphorylated combinatorially at positions 3, 4, and 5 to generate a set of seven PIs. These PIs are key regulators of sub-cellular processes; for example, phosphatidylinositol 4,5 bisphosphate [PI(4,5)P$_2$] plays a key role in the regulation of membrane transport and the cytoskeleton (*Janmey et al., 2018*), whereas phosphatidylinositol three phosphate (PI3P) is an important regulator of endocytosis and autophagy (*Schink et al., 2016*). PIs have also been implicated in the regulation of nuclear function (*Fiume et al., 2015*). PIs also play a key role in regulating developmental processes in metazoans. For example, phosphatidylinositol 3,4,5 trisphosphate (PIP$_3$) is an essential regulator of growth factor signaling and plays a conserved role in developmental control in *C. elegans*, *Drosophila*, and mammals (*Engelman et al., 2006*). Phosphatidylinositol four phosphate (PI4P) levels play a key role during cellular development in gametogenesis in *Drosophila* (*Brill et al., 2000*; *Tan et al., 2014*) and a conserved role for PI(4,5)P$_2$ is proposed in apico-basal polarity (*Devergne et al., 2014*; *Martin-Belmonte et al., 2007*; *Rousso et al., 2013*; *Yan et al., 2011*). Dysregulation in PI signaling has also been linked to several human diseases; these include developmental disorders such as Lowe syndrome (*Staiano et al., 2015*) and PIK3CA-related overgrowth spectrum (PROS)(*Madsen et al., 2018*); mutations in PTEN, a key regulator of PIP$_3$ levels are the second most frequent mutations seen in human cancers (*Worby and Dixon, 2014*). Mutations in genes regulating PI signaling are seen in a large number of genetic disorders of the human nervous system (*Raghu et al., 2019*). Thus, the PI signaling pathway is a key regulator of cell function and tissue architecture both during normal development and also in disease mechanisms and understanding their role in such processes is of fundamental importance.

Given their importance in a large number of important subcellular processes, PI signaling is tightly regulated. The seven PIs are generated in cells by the activity of a set of evolutionarily conserved lipid kinases that are specific for the substrate they will use, as well as the position on the inositol at which they will phosphorylate (*Sasaki et al., 2009*). In turn, once generated, PIs exert their cellular effects by binding to and modulating the activity of a large number of PI-binding proteins whose functions are key to the control of subcellular processes by these lipids. For example, many proteins contain domains that bind PI3P (PX and FYVE) and PI4P (FAPP and OSBP) and numerous PH domains that bind various PIs with differing degrees of affinity and selectivity have been reported (*Hammond and Balla, 2015*). Indeed, a recent biochemical study has identified a large number of proteins that bind a range of PIs with varying degrees of specificity (*Jungmichel et al., 2014*). However, the biological function of many of these proteins remains to be discovered.

Although PI signaling is broadly a conserved feature of all eukaryotic cells, some aspects of this pathway are found uniquely in metazoa and play important roles in animal development and tissue architecture. However, there remain key challenges in the analysis of PI signaling in metazoan development. For example, PIs may regulate the same subcellular process (e.g endocytosis) but may be co-opted to control the trafficking of unique cargoes in different tissues. Second, through their ability to control the secretion of intercellular signaling morphogens and hormones that mediate intercellular signaling, PIs can regulate tissue development in a non-cell autonomous manner. Third, given their key functions in cell biology, whole body knockouts of these genes often result in organismal lethality. Finally, in many cases, a given biochemical activity is underpinned by multiple genes in mammalian genomes and gene redundancy usually makes analysis of *in vivo* function challenging.

Analysis in *Drosophila* offers a powerful alternative to overcome the challenges of studying PI function in cell and developmental processes. Many developmental mechanisms are conserved between *Drosophila* and other metazoan models. A detailed annotation of the PI signaling genes in the *Drosophila* genome (*Balakrishnan et al., 2015*) led to the identification of 103 genes which are part of the PI signaling toolkit (*Figure 1*) and conserved between flies and other organisms important for the study of developmental processes and disease biology. Further the availability of binary gene expression control systems such as the GAL4/UAS system allow for tight spatial and temporal gene modulation in turn allowing sophisticated analysis of both cell autonomous and non-cell autonomous modes of developmental control by PI signaling (*Colombani et al., 2005*). Thus, *Drosophila* offers a model where one can analyze the control of cellular and developmental processes by PI signaling in a metazoan context.

In the recent years, CRISPR/Cas9 has emerged as a powerful tool for genome editing in multiple organisms including flies (*Gratz et al., 2013*; *Wang et al., 2016*; *Wu et al., 2018*). However, unlike transgenic fly lines expressing RNAi constructs that are available against almost the entire fly genome (*Dietzl et al., 2007*; *Perkins et al., 2015*), efforts to make guide RNA (gRNA) expressing transgenic flies are still underway (*Meltzer et al., 2019*; *Port et al., 2020*). In this study, we have generated a genome editing system that can be used for Cas9 mediated editing to delete the open-reading frame of each of the 103 annotated genes that could regulate PI signaling (*Balakrishnan et al., 2015*; *Supplementary file 1*). We demonstrate the use of this resource both in cultured *Drosophila* cells as well as in intact *Drosophila* tissues in vivo during development. Importantly, we use these guides to generate both whole body knock outs as well as tissue specific, developmentally timed depletion of individual genes with high efficiency. Further, using these reagents we present the results of a genetic screen in the developing *Drosophila* eye and identify key roles for PI signaling in this influential model of tissue development and patterning. Our reagent set will be a powerful and versatile tool for the discovery of novel regulation by PI signaling of development, tissue homeostasis, and human disease biology (*Supplementary file 1*).

## Results

### Design of gRNAs for editing PI signaling genes

In designing the gRNAs to target each PI signaling gene, we considered a few important factors: Firstly, CRISPR-Cas9-based genome editing is highly robust and specific; a single mismatch in the target sequence may highly reduce the efficiency of CRISPR (*Pattanayak et al., 2013*), especially if

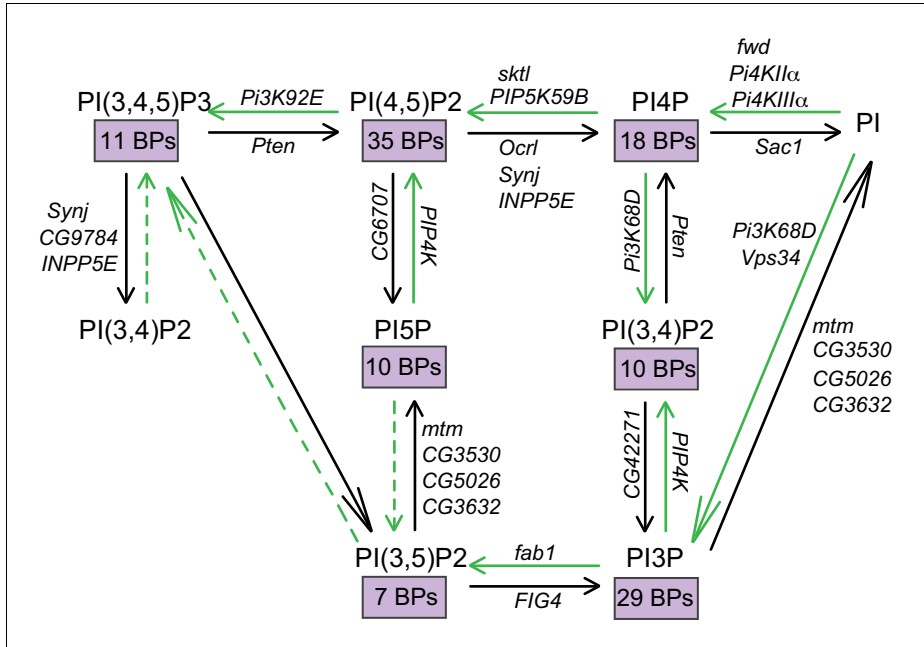

**Figure 1.** Interconversion of phosphoinositides in eukaryotic cells. The metabolic pathway by which the different phosphorylated forms of phosphatidylinositol are interconverted is represented. The kinase and phosphatase reactions are indicated by green and black arrows, respectively. The *Drosophila* gene encoding each of the enzymes responsible for these reactions are also indicated. Dotted lines indicate reactions that are yet to be established. The number of binding proteins (BPs) capable of binding each of the phosphoinositides are indicated in purple boxes.

the mismatches are present in the core 12 bp protospacer region most proximal to the protospacer adjacent motif (PAM) site. Second, a large number of users may find the library of these dual guide RNA (dgRNA) transgenic flies useful to study PI signaling in their own research contexts. This would mean that the dgRNA lines may be used in flies of different genetic backgrounds. Third, the same dgRNA constructs should be able to target the genes in S2R$^+$ cells for editing prior to biochemical experiments. Considering these points, we decided not to directly use the reference genome sequence to design the gRNAs. The entire genome of S2R$^+$ cells as well as the isogenized *attP40* parent fly stock (BDSC 25709: *y$^1$ v$^1$ P{y[+t7.7]=nos-phiC31\int.NLS}X; P{y[+t7.7]=CaryP}attP40*) that was to be used to generate dgRNA transgenic flies were sequenced (sequencing data is available for S2R$^+$ cells on hyperlink and for isogenized *attP40* stock on hyperlink). Sequences thus obtained were aligned against the reference genome and also against each other. When aligned against the reference genome, both the isogenized parent fly stock and the S2R$^+$ cell genome sequences showed more than 600,000 single-nucleotide polymorphisms (SNPs) (***Supplementary file 3***, ***4***). When compared to each other, these genome sequences had ~1.3 million mismatches (SNP, insertions, and deletions) suggesting that, as expected, almost all the sequence differences in the two genomes were independent of each other (***Supplementary file 3***, ***4***). While designing the single gRNA (sgRNA), we only chose those targets that did not have sequence mismatches in any of the three genomes (reference genome and the two genomes sequenced) thus ensuring that the dgRNAs could be used in flies of diverse genetic backgrounds and also in S2R$^+$ cells. Finally, the designed gRNAs were matched to recently added genomic sequences of reference genome r_6, *vasa-Cas9* (BDSC 51324: *w$^{1118}$; PBac{vas-Cas9}VK00027*), *nos-Cas9 II* (BDSC 78781: *y$^1$ sc$^*$ v$^1$ sev$^{21}$; P{nos-Cas9. R}attP40*) and *nos-Cas9 III* (BDSC 78782: *y$^1$ sc$^*$ v$^1$ sev$^{21}$; P{nos-Cas9.R}attP2*). Only 1 gRNA target (targeting *CG10426*), out of the 206 generated, seemed to have one base mismatch marked in red (***Supplementary file 1***), while the other 205 targets matched between all the genomes. These gRNA can hence be used in many genetic backgrounds. This allowed for a relatively simple way to test the designed dgRNAs in S2R$^+$ cells for their ability to target specific genes prior to generating transgenic fly lines.

For each gene, two regions were identified wherein an sgRNA would be designed to target each region. The first sgRNA would be designed to target the first coding exon and the second sgRNA to target the exon with the stop codon (*Figure 2*). In case of genes with alternate spliced forms, regions around the most 5' start codon and the most 3' stop codon were chosen (For example, *CG3682* has upto six splice variants with four putative start sites). In cases where two or more putative genes are annotated on the same locus, sgRNA target regions were chosen such that the coding sequences of neighboring genes are not disrupted (for example, there are six different putative ORFs in the first few introns of *plc21c* and hence the first sgRNA is designed on exon 8). Once a set of two ~ 100 bp regions were identified for each gene keeping in mind the above-mentioned criteria, we designed the sgRNAs using an online tool (http://targetfinder.flycrispr.neuro.brown.edu) with zero predicted off-targets. During the course of this study, an additional online tool (https://www.flyrnai.org/evaluateCrispr/) became available. Thereafter, this tool was used to obtain a predicted efficiency score. The most efficient sgRNAs in the region were then checked in our sequence database for presence of any mismatches in either S2R$^+$ cells or injection fly stocks compared to the reference genome sequence. The best sgRNA sequences predicted to efficiently target specific genes with no predicted off-targets and absence of any mismatches were chosen for synthesis. Any sgRNA sequences that did not qualify these criteria were discarded and new sgRNAs designed to fit all the above criteria. A full list of genes for which dgRNAs were designed is listed in *Supplementary file 1*.

## Generation of gRNA expression constructs

The first sgRNAs targeting a region close to the start codon for each gene were cloned into the pBFv6.2 vector and the second sgRNAs targeting a region close to the stop codon into the pBFv6.2B vector (see Materials and methods for details). Both these vectors ubiquitously express the gRNAs under the U6.2 promoter (*Kondo and Ueda, 2013*). The two sgRNA constructs targeting a given gene along with hsp70-Cas9 were co-transfected into S2R$^+$ cells. The sgRNA pairs that led to the deletion of the gene, as tested by PCR and sequencing of the target gene loci, were chosen for dual-gRNA (dgRNA) construction. In case a particular pair of sgRNAs failed to delete the target gene, a new set of sgRNAs were designed for both the first and the last coding exon and different combinations of sgRNAs were tested until the most optimal pair capable of deleting the target gene was identified.

Once a functional pair of sgRNAs capable of deleting a target gene was identified, the first sgRNA was cloned into the pBFv6.2B vector containing the second sgRNA to generate the dgRNA construct (See Materials and methods for details). These constructs were then microinjected to generate dgRNA transgenic flies for each of the 103 PI signaling genes (*Figure 2A*). This dgRNA construct can also be transfected in S2R$^+$ cells to generate gene deletions for cell culture-based experiments. For genes that are present on the second chromosome, the dgRNA construct was inserted on the third chromosome (*Supplementary file 1*). For all the other genes, the dgRNA was inserted on the second chromosome (See Materials and methods). This design offers the advantage that when used to generate whole fly knockouts, the dgRNA transgenes can subsequently be readily out-crossed given that the gRNA and the target gene are on two different chromosomes. Once generated, all the dgRNA transgenics were crossed to *Act5c-Cas9* (BDSC 54590: $y^1$ M{Act5C-Cas9.P} ZH-2A $w^*$) flies and larval progeny were tested for deletions by PCR and sequencing. Out of 103 genes, 102 gave the desired deletion product, while one did not give a band of the expected length (*Supplementary file 1*). This could be either because this gRNA pair is not functional in flies, or the PCR or deletions were of low efficiency.

To test whether protein levels were reduced significantly when the target locus was removed, we stained larval nephrocytes with an antibody to the OCRL protein (*Del Signore et al., 2017*). To test the specificity of the antibody, we used a null germline mutant of *ocrl* (*ocrl^KO^*) generated in this study (see sections below) that removes the open-reading frame of *ocrl*. In western blots from wild-type animals, a band of 97 kDa was detected that was absent in *ocrl^KO^* (*Figure 2B*). Immunocytochemistry of larval nephrocytes revealed punctate staining throughout the cell body that was completely absent in nephrocytes of *ocrl^KO^* (*Figure 2C*). We used transgenic flies expressing the guides used to generate *ocrl^KO^* but the Cas9, was expressed only in nephrocytes using *Dot Gal4.* The expression of Cas9 protein was verified by visualizing GFP fluorescence (described in section below) in nephrocytes (*Figure 2E*) and the deletion of *ocrl* was verified using primers that specifically detect the deletion product. We obtained an amplicon of 555 bp

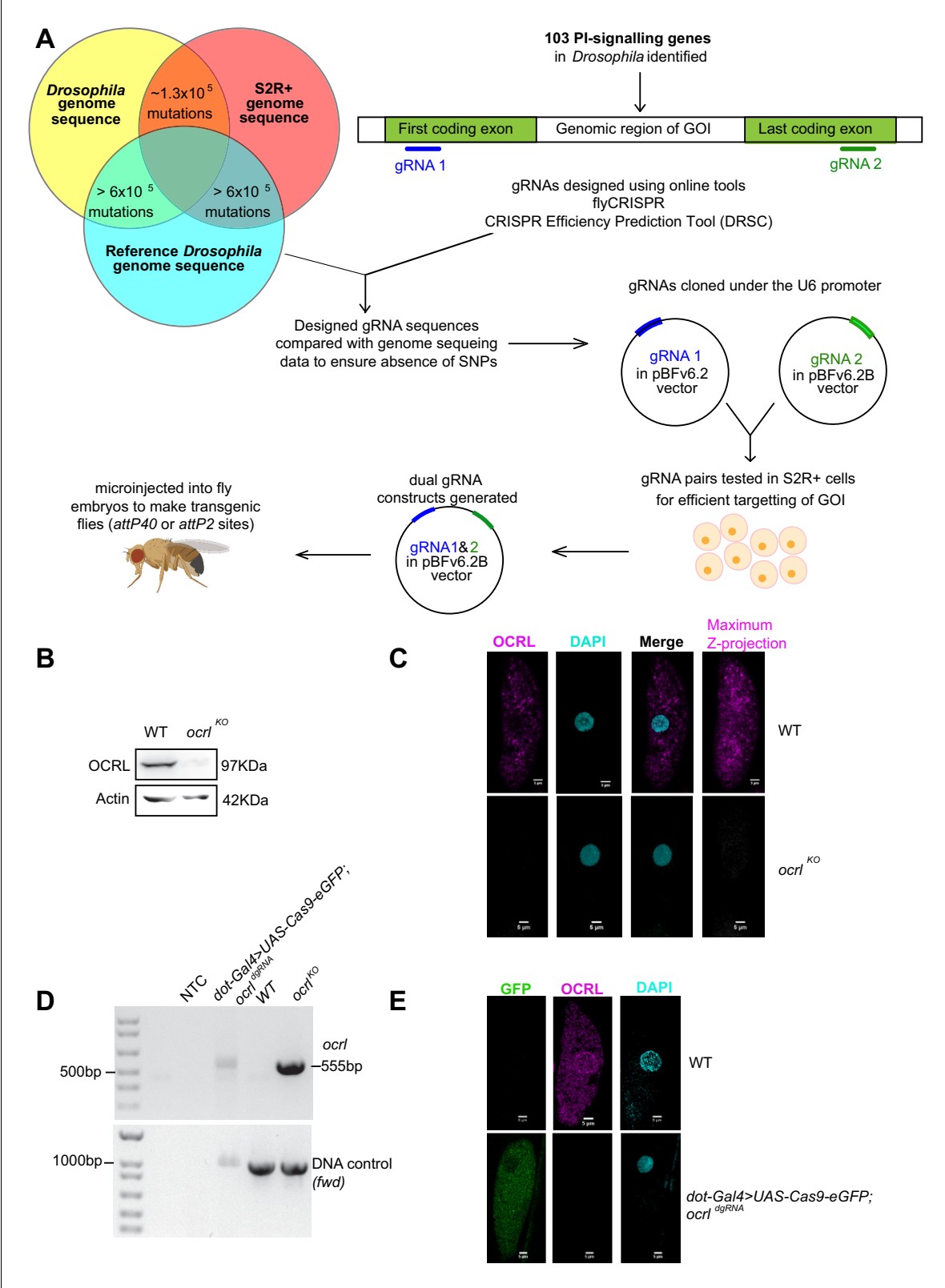

**Figure 2.** Generation of dgRNA targeting PI signaling genes. (**A**) Workflow for the generation of dgRNA transgenic flies. For each of the 103 PI signaling genes in the *Drosophila* genome, two gRNAs were designed using flyCRISPR and verified using the CRISPR Efficiency Prediction Tool (DRSC). The first gRNA (gRNA 1; indicated in blue) was designed to target the first coding exon and the second gRNA (gRNA 2; indicated in green) was designed to target the last coding exon. It was ensured that these designed gRNAs did not have any mismatches when compared against any of the

*Figure 2 continued on next page*

*Figure 2 continued*

three genomes (the genomes of *Drosophila* BL25709 line used for microinjection, S2R+ cells and the reference genome). The two gRNAs (gRNA one and gRNA 2) for each gene were cloned into pBFv6.2 and pBFv6.2B, respectively, and tested in S2R+ cells for their ability to delete the target genes in the presence of Cas9. Following this, both the gRNAs for each gene were cloned into a single plasmid to generate the dgRNA constructs. These were microinjected into *Drosophila* embryos to generate dgRNA transgenic flies against each of the PI-signaling genes. (B) Western blot from protein lysates of 3rd instar larvae from wild type and *ocrl*KO. The OCRL polypeptide of the expected Mr of ca. 100 kDa is completely absent in *ocrl*KO lysates. Protein levels of Actin are used as the loading control. OCRL antibody used here has been described in *Del Signore et al., 2017*. (C) Immunolabeling of pericardial nephrocytes from 3rd instar larvae to examine the distribution of the OCRL protein. As expected, the OCRL protein (magenta) is seen in punctate structures throughout the cell body. This staining is completely absent in *ocrl*KO animals. The experiment was repeated three times and multiple animals examined in each case. A single nephrocyte is shown for illustrative purposes. (D) PCR analysis demonstrating the 555 bp band detected when *ocrl* is edited by these dgRNAs is shown. For nephrocyte-specific deletion, third instar larvae were dissected and the body wall (including heart tube and nephrocytes) was used for the DNA preparation. As a positive control, the germ line *ocrl*KO larval lysates were used. *CG7004* (*fwd*) is used as a control gene for the quality and quantity of the DNA prep. The 555 bp diagnostic band is seen in the *ocrl* KO and nephrocyte-specific knockout DNA but not in wild-type flies and in NTC (no template control). (E) Immunostaining of nephrocytes in nephrocyte-specific knockout of *ocrl*. A single nephrocyte is shown. Green channel shows the GFP generated from the Cas9-eGFP transgene. Magenta channel shows immunolabeling with the OCRL antibody. Nucleus is stained with DAPI. There is no detectable signal for OCRL in the nephrocytes of nephrocyte specific knockout animals; the experiment was repeated twice and 24 individual nephrocytes images from multiple animals.

confirming the deletion of *ocrl* in nephrocytes (*Figure 2D*); no deletion band was found in control animals, confirming nephrocyte-specific deletion of *ocrl*. Immunolabeling with an antibody to OCRL confirmed the absence of staining for this protein in the nephrocytes of these larvae with nephrocyte-specific deletion of *ocrl* (*Figure 2E*).

## Generation of *UAS-Cas9 -T2A-eGFP* transgenic lines

At the core of CRISPR/Cas9 technology is the endonuclease Cas9 which utilizes gRNAs to target specific genomic sequences and generate a double-stranded break. Transgenic flies expressing Cas9 under UAS control are already available (*Port et al., 2020*). However, these lack a reporter and therefore it is difficult to readily identify cells or tissues expressing Cas9, which would be of great advantage when targeting genes in a tissue-specific manner. In order to satisfy this requirement, we designed and generated transgenic *UAS-Cas9* flies with a fluorescence reporter eGFP. Given that the human codon optimized Cas9 (Cas9.P2) expresses at lower levels thereby reducing the cytotoxic effects of the otherwise highly expressing Cas9.C (*Meltzer et al., 2019*), we have used Cas9.P2 to generate our construct. In order to ensure that the presence of an eGFP tag does not hinder the activity of Cas9, we introduced a self-cleaving peptide T2A sequence in between Cas9 and eGFP (*Figure 3A*). This ensured that while the Cas9 and eGFP are expressed from the same mRNA, they are made as two independent proteins. Presence of the eGFP coding sequence downstream of Cas9 ensured that every cell positive for eGFP fluorescence would definitely also express Cas9. In addition, we tagged the Cas9 with a nuclear localization signal (NLS) at both the N- and the C-terminal to facilitate its translocation into the nucleus for better access of the genomic DNA.

The *UAS-Cas9-T2A-eGFP* construct thus generated was tested for its genome editing efficiency and the usefulness of eGFP as a reporter. To test the Cas9, the *UAS-Cas9-T2A-eGFP* construct was transfected along with the dgRNA against *CG5734* (a gene predicted to have a PH domain that may bind to PI3P) into S2R+ cells constitutively expressing *Tubulin-Gal4*. As a control, the cells were parallelly co-transfected with the pBS-hsp70-Cas9 plasmid (Addgene Plasmid #46294) and the dgRNA against *CG5734*. Forty-eight hours post-transfection the cells were harvested. Half the cells were used for protein extraction and western blotting. Western blotting showed that Cas9 was being expressed from the *UAS-Cas9-T2A-eGFP* construct. The molecular size of Cas9 from this construct was similar to Cas9 expressed from *pBS-hsp70-Cas9*. Detection of a band corresponding to the molecular weight of free eGFP in cells transfected with *UAS-Cas9-T2A-eGFP* suggested that the T2A sequence was working efficiently to generate Cas9 and eGFP as two independent proteins from a single mRNA (*Figure 3B*). From the other half of the cells, genomic DNA was isolated and PCR performed to detect deletions in *CG5734*. We found the predicted deletion fragment generated by both the Cas9 from the newly generated *UAS-Cas9-T2A-eGFP* plasmid and the Cas9 expressed from the *pBS-hsp70-Cas9* control plasmid (*Figure 3C*) thus verifying that Cas9 expressed from the

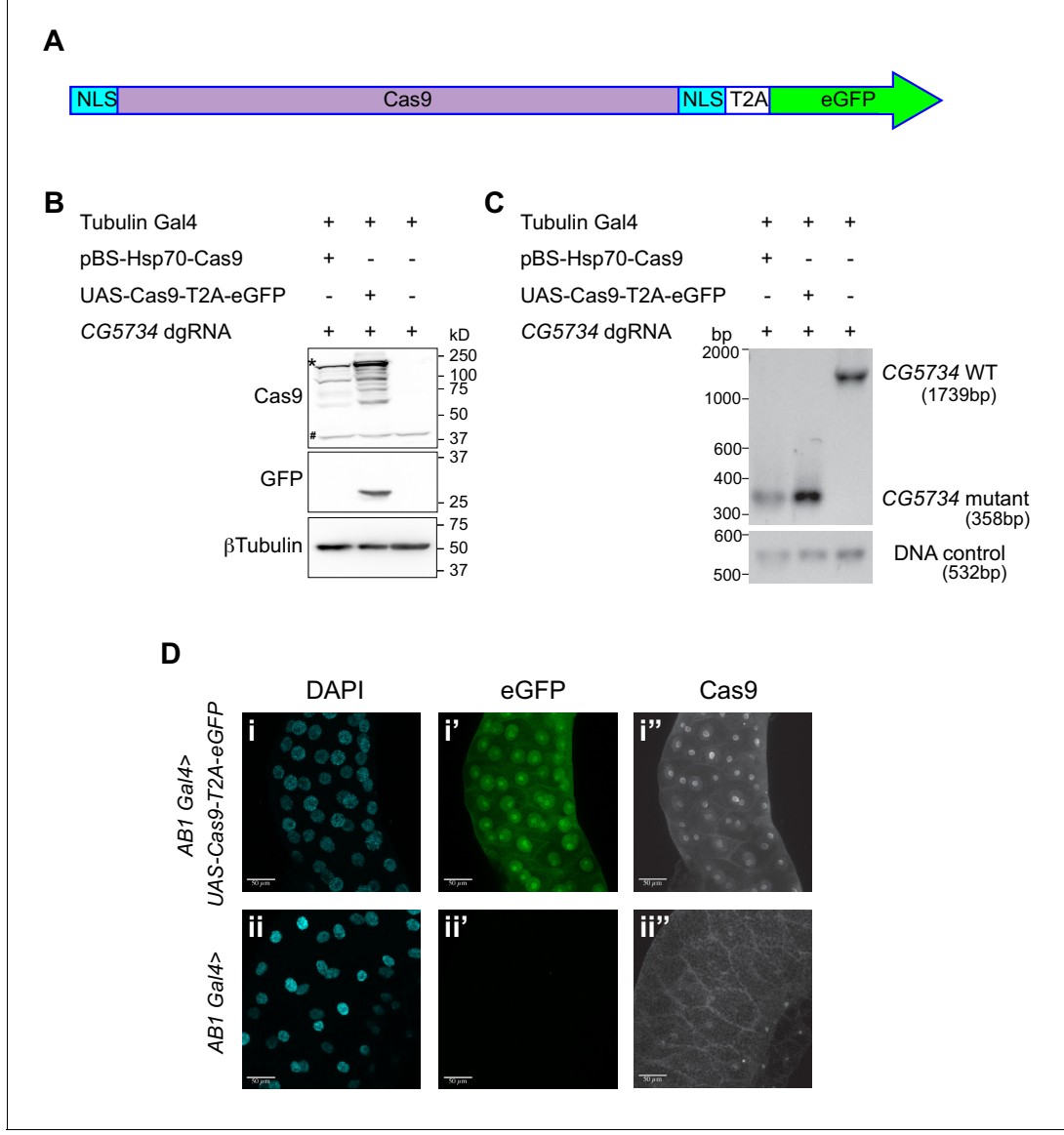

**Figure 3.** Design and validation of *UAS-Cas9-T2A-eGFP*. (**A**) A schematic of the *Cas9-T2A-eGFP* transgene indicating the presence of a nuclear localization signal (NLS-blue box) at the N and C termini of Cas9; Cas9 is shaded in pink. An eGFP sequence (green) is present downstream of the Cas9 sequence and these two are separated by the T2A sequence (T2A white box). (**B**) Western blot of S2R$^+$ cells expressing Cas9-T2A-eGFP. S2R$^+$ cells constitutively expressing Tubulin Gal4 were transiently transfected with the indicated plasmids. Cells were harvested 48 hr after transfection and one half of the cells were subjected to western blot analysis to verify that the T2A sequence was efficient and therefore resulted in expression of Cas9 and eGFP as independent proteins. The expected molecular size of Cas9 is 158 kD (the band of highest intensity indicated with a '*'). Note that the Cas9 antibody cross-reacts with a protein from S2R$^+$ cells ('#') thereby precluding immunocytochemical detection of Cas9 in S2R$^+$ cells. (**C**) Genomic PCR was performed on the other half of the cells to test for the ability of Cas9 to target *CG5734*. The presence of a 358 bp band in cells expressing Cas9 compared to the 1739 bp band in untransfected cells suggests the successful deletion of *CG5734*. (**D**) *UAS-Cas9-T2A-eGFP* transgenic flies generated were crossed to *AB1-Gal4* flies and the salivary glands of the progeny dissected and stained using Cas9 antibody (1:200). Cas9 (grayscale) was predominantly localized in the nucleus suggesting efficient nuclear localization by the NLS. Cells expressing Cas9 were marked by eGFP. Scale bar is 50 μm.

*UAS-Cas9-T2A-eGFP* construct was capable of deleting target genes in the presence of appropriate gRNAs.

After verification in S2R$^+$ cells, the *UAS-Cas9-T2A-eGFP* construct was microinjected into fly embryos along with a helper plasmid expressing transposase to facilitate random P-element-based insertion and obtain transgenic flies. In order to verify that the fly line obtained expresses Cas9, we

crossed these lines to salivary-gland-specific Gal4 (*AB1-Gal4*) flies. The progeny flies were dissected, salivary glands stained for Cas9 and imaged. All salivary gland cells expressed eGFP and were also stained positive for Cas9 thus demonstrating that the *UAS-Cas9-T2A-eGFP* construct can be used as expected to drive Cas9 expression and to mark the Cas9-expressing cells with eGFP (*Figure 3D*). The eGFP localization was predominantly nuclear although some cytosolic localization could be observed. Cas9 did not have the same intracellular localization thus suggesting that the T2A sequence was functional and that the two proteins, Cas9 and eGFP were being expressed as independent proteins. Owing to the NLSs attached to the Cas9, its expression was limited to the nucleus. The *UAS-Cas9-T2A-eGFP* flies showed Cas9 expression similar to the already existing Cas9 transgenic flies (BDSC 54592: *P{ry[+t7.2]=hsFLP}[1], y[1] w[1118]; P{y[+t7.7] w[+mC]=UAS-Cas9.P}attP2/ TM6B, Tb[1]*) with the added advantage of eGFP expression to mark Cas9-expressing cells. These *UAS-Cas9-T2A-eGFP* flies were used for all experiments described further.

## Investigating the role of PI signaling in eye development using the dgRNA toolkit

The *Drosophila* eye serves as an excellent model to perform genetic screens as it is not essential for viability and offers a phenotype that can be easily scored. Previously, influential studies have used *Drosophila* eyes to screen for genes regulating cell growth and proliferation [reviewed in *Tseng and Hariharan, 2002*]. The *Drosophila* eye is composed of ~800 regularly arranged ommatidia, each containing eight photoreceptor neurons making the *Drosophila* eye an attractive tissue to study neurodegenerative diseases like Huntington's disease (*Krench and Littleton, 2013*) and screen for genes and genetic modifiers involved in neural development (*Ma et al., 2009*). Moreover, photoreceptor neurons are an excellent model for phototransduction which in flies is largely dependent on PI signaling (*Raghu et al., 2012*) and dysregulation of PI signaling leads to defects in the development, cellular structure and functions of the eye.

To test the use of our dgRNA transgenic library, we studied the effect of deletion of PI signaling genes in the eye disc during development by expressing *Cas9-T2A-eGFP* using *eyeless-Gal4* (*eyGal4*). *eyeless (ey)* is expressed very early on in the eye primordia in the embryo and before the formation of the morphogenetic furrow at the time of photoreceptor determination in the third instar larva (*Halder et al., 1995*). In addition, *ey* is also expressed in neurons (*Callaerts et al., 2001*). To delete target genes in a tissue specific manner we generated *ey-Gal4; UAS-Cas9-T2A-eGFP* flies and crossed them individually to each of the 103 *dgRNA* transgenic lines. Although we know that these gRNAs are functional in flies (*as* seen by PCR and sequencing in gRNAs crossed to *Act5c-Cas9- Supplementary file 1*), we wanted to confirm whether the gRNAs were functional in the eye specific context. Hence, prior to analyzing the phenotypes obtained in the progeny of these crosses, we tested whether the gene under study was being edited in a tissue-specific manner. For this, we isolated genomic DNA from the head and the body of a few representative *ey >GAL4; UAS- Cas9-T2A-eGFP; dgRNA* flies and performed a PCR to detect the expected genomic deletion. Expected genomic deletions were identified in DNA obtained from fly heads but not in DNA obtained from the fly body suggesting tissue (eye)-specific gene deletion using our newly generated reagents (*Figure 4A*). This suggested that the *UAS-Cas9-T2A-eGFP* was indeed under the control of *eyGal4* and had minimal or no influence from other genetic elements despite being generated by random P-element insertion. In most cases, no wild-type band was obtained where gRNA was used (*Figure 4A*), either because the mutation was highly efficient, or more likely because the smaller PCR products are more likely to be amplified. This is consistent with absence of WT bands seen in the mosaic F1 progeny from cross between *Act5c-Cas9* and *Dp110[dgRNA]* (*Figure 4—figure supplement 1*). However, we did get a small amount of wild-type products in some cases (*Figure 4—figure supplement 1 - clc*).

The eyes of progeny flies from each of the crosses between *ey-Gal4; UAS-Cas9-T2A-eGFP* and the 103 *dgRNA* transgenic lines were imaged with appropriate controls. A spectrum of phenotypes was obtained ranging from smaller eyes as in the case of *Dp110* [*Figure 4B* (i)], presence of necrotic patches like with *Vps34* [*Figure 4B* (ii)] and larger eyes when *Pten* was knocked out [*Figure 4C* (i)]. *Sac1* knockout led to smaller eyes that were crumbled [*Figure 4C* (ii)]. CRISPR-mediated knockout of PI binding proteins *Abl* [*Figure 4D* (i)] and *Clc* [*Figure 4D* (ii)] resulted in an irregular arrangement of ommatidia giving a rough appearance to the eyes. This suggested possible different roles for these proteins during eye development. In some cases,

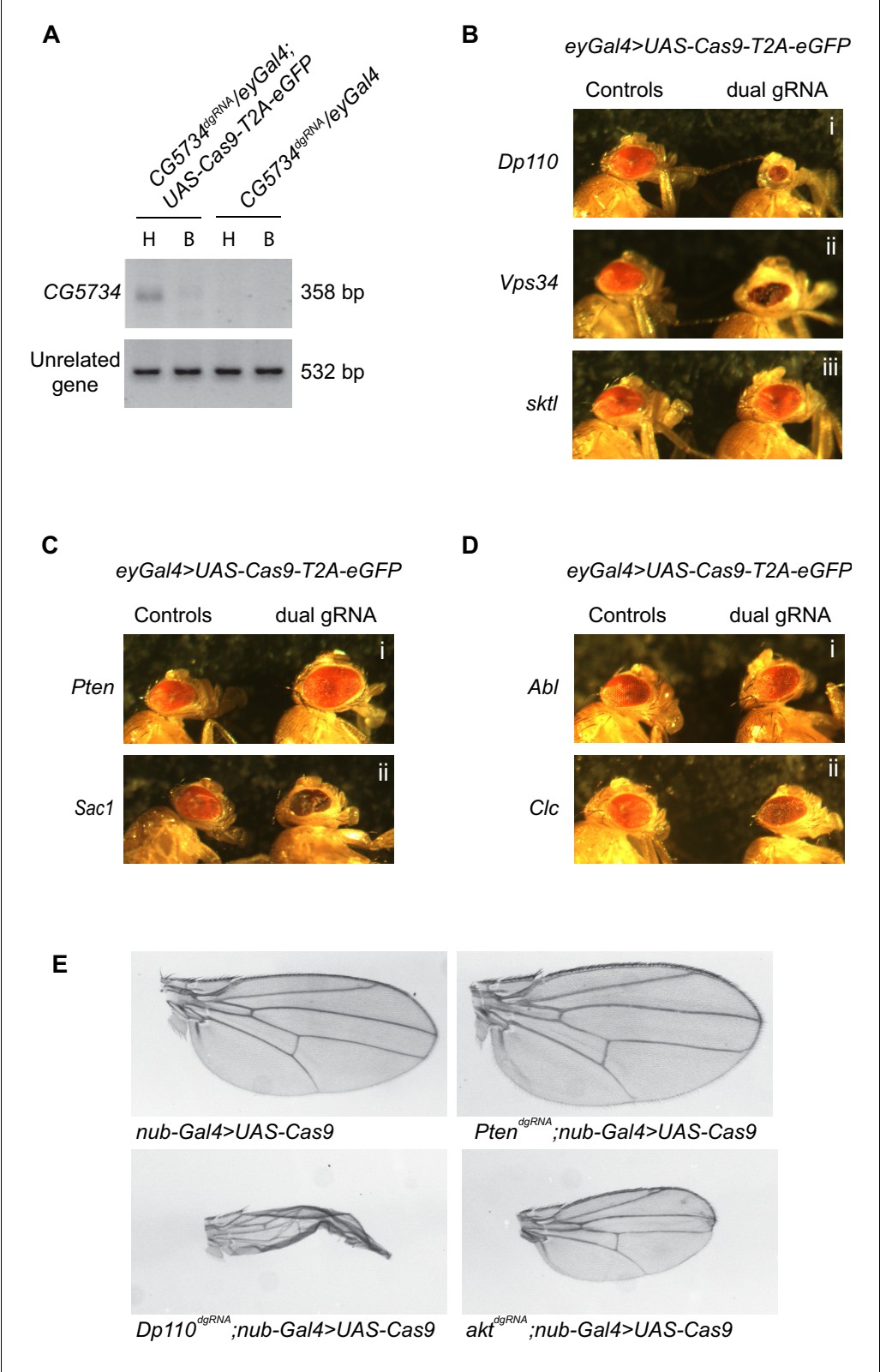

**Figure 4.** A genetic screen employing the *dgRNA* transgenic fly library to identify PI signaling genes in eye development. (**A**) Genomic PCR of *CG5734* as a representative example to show that tissue specific gene deletion can be obtained by driving *UAS-Cas9-T2A-eGFP* specifically in the eyes using eyGal4 in the presence of ubiquitously expressing dgRNAs. An amplicon corresponding to *CG5734* deletion (See **Supplementary file 1**) was obtained in DNA extracted from fly heads ('H') of the appropriate genotype indicated but not in DNA extracted from the bodies ('B') of the fly. (**B**) *Figure 4 continued on next page*

Figure 4 continued

Representative images of phenotypes observed upon eye specific CRISPR mediated deletion of a few PI kinases – *Dp110, Vps34* and *sktl*.
(C) Representative images of phenotypes observed upon eye-specific CRISPR-mediated deletion of two PI phosphatases – *Pten* and *Sac1*. (D)
Representative images of phenotypes observed upon eye-specific CRISPR-mediated deletion of PI binding proteins *Abl* and *Clc*. (E) *Nub-Gal4;UAS-
Cas9* flies (BDSC 67086: *w\*; P{GawB}nubbin-AC-62; P{UAS-Cas9.P2}attP2*) were crossed to different genotypes and clipped wings of the progeny were
imaged. Number of flies inspected per genotype is given in parenthesis: *Nub-Gal4/+;UAS-Cas9/+* (17), *Pten$^{dgRNA}$/Nub-Gal4;UAS-Cas9/+* (15),
*Dp110$^{dgRNA}$/Nub-Gal4;UAS-Cas9/+* (6) and *akt$^{dgRNA}$/Nub-Gal4;UAS-Cas9/+* (12).
The online version of this article includes the following figure supplement(s) for figure 4:

**Figure supplement 1.** Genomic deletion using Gal4/UAS system.
**Figure supplement 2.** Genomic deletion obtained in CRISPR tissue-specific KOs is comparable irrespective of the presence or absence of phenotype.

For example *skittles* [*Figure 4B* (iii)], eye morphology was comparable to the control flies with no visible morphological differences. To confirm that the lack of phenotype was not due to lower efficiency of deletion at the target locus, we performed deletion PCR for representative target loci where either a phenotype was observed or not observed (*Figure 4—figure supplement 2*). It was observed that target locus deletion was seen irrespective of presence or absence of phenotype. Thus, the absence of a phenotype when some genes are deleted is not a consequence of the lack of a deletion; rather these may be examples where the protein pre-existing in the precursor cells perdures despite the gene being subsequently deleted. The details of all the 103 genes and phenotypes obtained have been listed in *Supplementary file 1*. In addition to studies in the developing eye, we also deleted a subset of genes that gave strong phenotypes in the eye in the developing wing disc using the domain-specific *nub-GAL4*. Deletions of *Pten*, *Dp110* and *akt* all resulted in the changes in wing size and morphology that were consistent with the known role of these genes (*Figure 4E*). Thus, deletion of *Pten* resulted in a wing blade larger than that of controls while deletion of *Dp110* and *akt* both resulted in a reduction in the size of the wing blade.

Deletion of Class I PI3K (*Dp110*), that converts PI(4,5)P$_2$ to PI(3,4,5)P$_3$ resulted in smaller eyes compared to controls. This is in agreement with a number of studies that have established the importance of PI3K for cell growth (*Goberdhan et al., 1999*; *Weinkove et al., 1999*). In order to compare the efficiency of using CRISPR-based knockout versus RNAi-mediated knockdown when studying tissue-specific phenotypes, we expressed the *Dp110$^{dgRNA}$* and two different *Dp110$^{RNAi}$* lines in the *Drosophila* eye using eyGal4. All three of these resulted in reduced size of the eye. While the RNAi-mediated knockdown resulted in ~40% reduction [*Figure 5A* (ii) and (iii) and C] compared to controls [*Figure 5A* (i)], dgRNA mediated *Dp110* knockout showed a stronger phenotype with eyes ~ 75% smaller [*Figure 5B* (ii) and C] than appropriate controls [*Figure 5B* (i)] thus demonstrating that these dgRNAs serve as efficient tools to study tissue-specific roles of genes.

## Targeting multiple genes using dgRNAs to study genetic interaction in specific tissues

Understanding genetic interactions is an important approach to establish the molecular pathways underpinning any given biological process. In order to demonstrate that the dgRNA library can be used to target multiple genes simultaneously in a tissue-specific manner, we took advantage of the reversible nature of the PIP$_2$ to PIP$_3$ conversion by enzymes *Dp110* and *Pten*. Class I PI3K (encoded by *Dp110*) phosphorylates PIP$_2$ to generate PIP$_3$ while the lipid phosphatase PTEN (encoded by *Pten*) dephosphorylates PIP$_3$ to generate PIP$_2$. We generated a strain containing both *Dp110$^{dgRNA}$* and *Pten$^{dgRNA}$* and crossed it to *ey-Gal4; UAS-Cas9-T2A-eGFP* flies. Disruption of *Pten* was able to rescue the phenotype of the disruption of *Dp110* and vice versa. The eyes of the *Dp110, Pten* double knockout flies were intermediate in size [*Figure 5B* (iv) and C] compared to the small eyes of the *Dp110* knockout [*Figure 5B* (ii)] and the large eyes of the *Pten* knockout flies (*Figure 5B* (iii) and C). However, the eyes of these double mutant flies were smaller compared to control flies possibly because the *Dp110$^{dgRNA}$* was more efficient than the *Pten$^{dgRNA}$* resulting in a greater reduction of PIP$_3$ levels.

The fact that multiple genes can be targeted simultaneously in a tissue-specific manner using CRISPR/Cas9 technology is of great advantage. Traditionally, genetic interactions have been studied by generation of double mutants through meiotic recombination. However, this may not be a viable

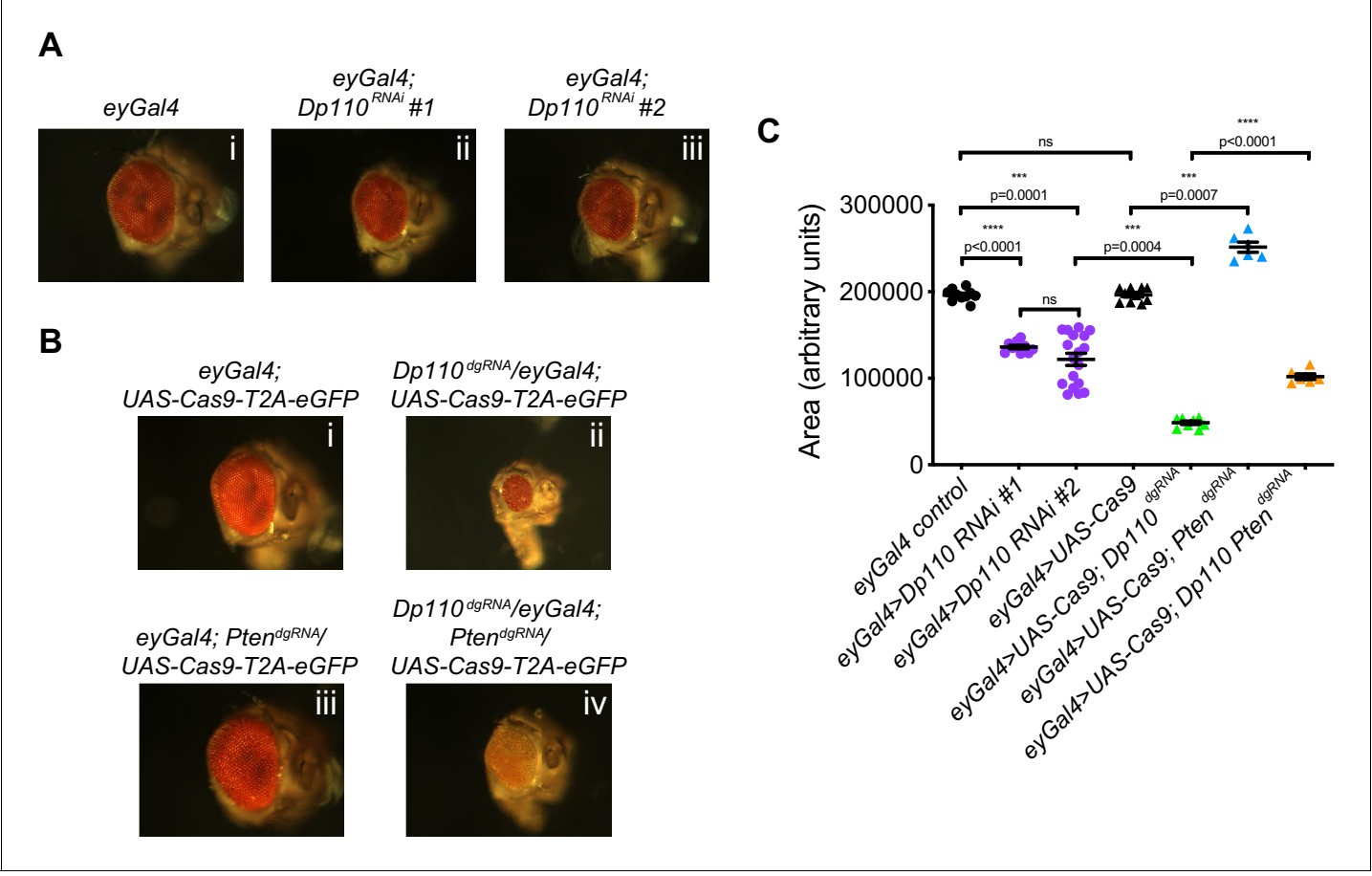

**Figure 5.** Combinatorial use of dgRNAs to study genetic interactions regulating developmental processes. (**A**) Knockdown of *Dp110* results in smaller eyes compared to control eyes as seen with two different RNAi lines (*Dp110$^{RNAi}$ #1*: BDSC 61182 and *Dp110$^{RNAi}$ #2*: BDSC 27690). (**B**) CRISPR mediated *Dp110* knockout phenocopies RNAi mediated *Dp110* knockdown and results in smaller eyes compared to control eyes. However, the phenotype observed with the use of CRISPR to target *Dp110* is more severe compared to the phenotype seen with either of the RNAi lines. CRISPR-mediated *Pten* deletion results in eyes larger than control eyes. When both *Dp110* and *Pten* are targeted simultaneously, the size of the eyes are intermediate between the small eyes seen with *Dp110* deletion and the large eyes seen with *Pten* deletion. (**C**) Quantifications of the phenotypes shown in A and B. Each data point is an individual eye. Each point on Y-axis represents area of eye ± s.e.m. (p values are indicated on the graph, ns = not significant; one-way ANOVA with Tukey's multiple comparison test).

option in many cases wherein the genes being studied lead to organismal lethality when disrupted or when the genetic loci of the two genes being studied are very close thereby drastically reducing the recombination efficiency between these genes. In fact, several genes involved in the PI signaling cascade are clustered together on different chromosomes. Our dgRNA transgenic fly library offers an opportunity to generate tissue-specific double knockouts independent of the proximity of these genes.

## Generation of whole fly knockouts using dgRNA transgenics

RDGB (retinal degeneration B) is a PI transfer protein (encoded by *rdgB*), depletion of which leads to light-dependent phototransduction defects (*Yadav et al., 2015*). Whole body mutants of *rdgB* are viable with no morphological defects in the eye (*Yadav et al., 2015*), although electrical recordings (ERG) from the eye of *rdgB* mutants show reduced amplitude. In order to rule out the possibility that the *rdgB$^{dgRNA}$* was non-functional in developing eye discs, we crossed the same *rdgB$^{dgRNA}$* transgenic flies to *nanos-Cas9* to generate a complete germline knockout of *rdgB* (*Figure 6A*). ERG recordings from this germline knockout of *rdgB* generated using CRISPR/Cas9 was comparable to *rdgB$^2$* (*Figure 6B*), an amorphic allele generated by chemical mutagenesis in the Benzer lab (*Harris and Stark, 1977*). In a similar manner, using the *dgRNA* transgenic flies, we generated

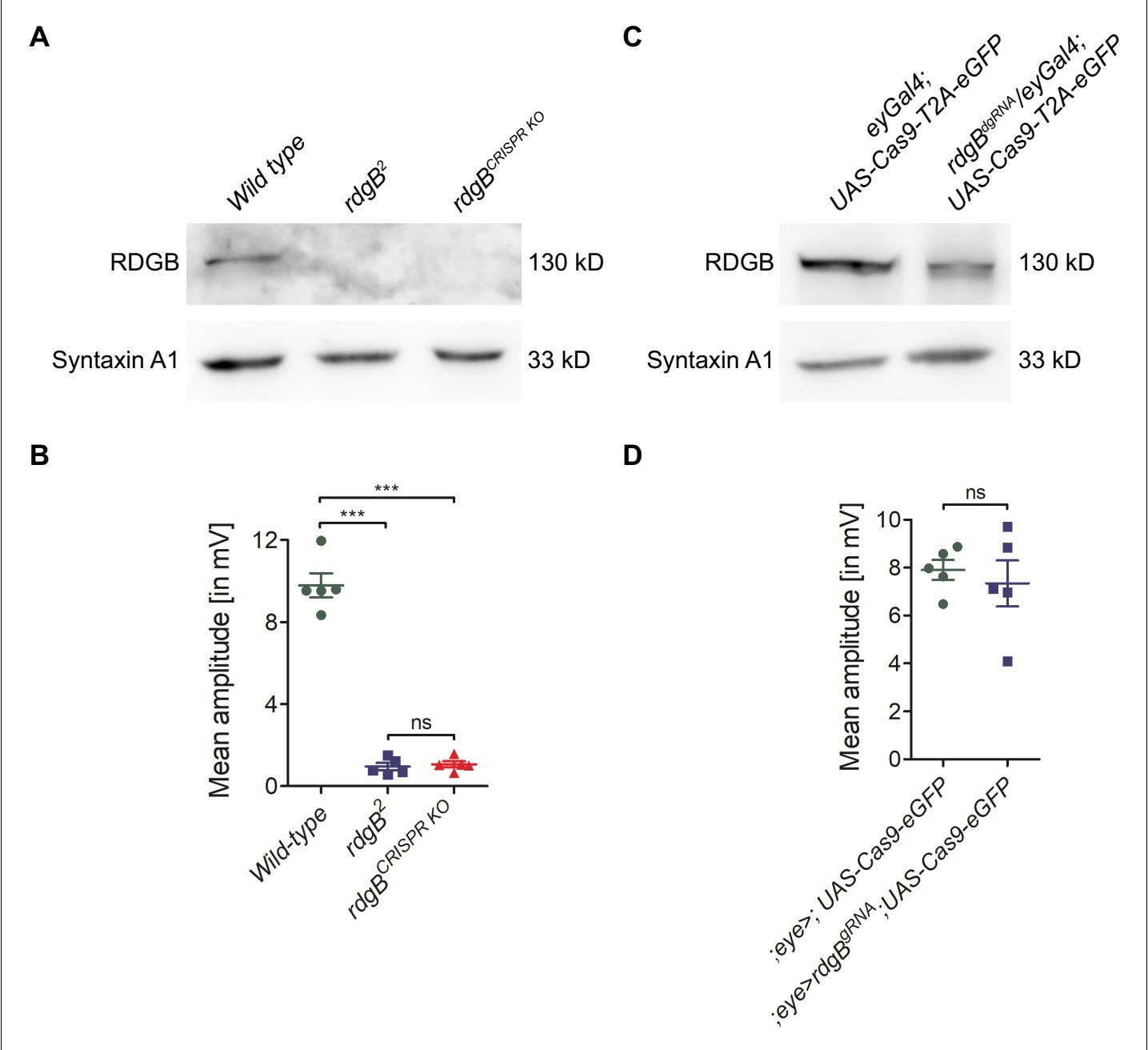

**Figure 6.** Generating germ line fly knockouts using dgRNA transgenics. The PI transfer protein RDGB (retinal degeneration B) is a good example to demonstrate the need for whole fly knockouts. (**A**) Western blot analysis showed that eye-specific Cas9 expression was not sufficient to knockout *rdgB* and detectable levels of RDGB protein was still present. (**B**) The mean amplitude from ERG recordings of these flies was the same as control flies. Each data point on the graph is recording from an individual eye. Each point on Y-axis represents mean amplitude ± s.e.m. (*** - $p<0.001$, ns = not significant; two tailed unpaired t-test). (**C**) The same dgRNA flies were used to generate a whole fly *rdgB* deletion (*rdgB*$^{CRISPR\ KO}$) and western blot analysis showed a complete loss of RDGB similar to the *rdgB*$^2$ null mutant flies. (**D**) Both *rdgB*$^2$ and *rdgB*$^{CRISPR\ KO}$ flies showed complete loss of ERG amplitudes as expected. Each point on Y-axis represents mean amplitude ± s.e.m. (*** - $p<0.001$, ns = not significant; two tailed unpaired t-test).

germline knockouts of several additional genes (total 48 genes) by crossing them to either *nanos*, *vasa or Act5c-Cas9* flies (**Supplementary file 1**). Several gRNAs when crossed to *Act5c-Cas9* or *vasa-Cas9* flies showed pupal lethality and hence were then crossed to *nanos-Cas9* flies. The deletions obtained in this manner are highly efficient between 4 and 100% efficiency in $F_1$ flies and between 3–70% efficiency in $F_2$ flies (absolute values provided in **Supplementary file 1**) in

generating germline knockouts with deletions ranging in size between 400 bp and 38 kb. For a few genes (13 genes), we were unable to generate germline deletions despite repeated attempts in some cases possibly due to their essential roles in gametogenesis or embryogenesis.

## Anomalous behavior of the Cas9 deletion system

We crossed the *rdgB*$^{dgRNA}$ flies with *ey-Gal4;UAS-Cas9-T2A-eGFP* flies. The progeny were collected and their ERG recorded. In contrast to the *rdgB*$^2$ allele and the germline *rdgB* knockout generated in this study (see previous section), we observed no reduction in ERG amplitude compared to wild-type flies (*Figure 6D*). Although PCR analysis of these eyes confirmed the occurrence of deletion events, western blot analysis from heads of *rdgB*$^{dgRNA}$, *ey-Gal4;UAS-Cas9-T2A-eGFP* flies revealed that the RDGB protein was still present in these eyes (*Figure 6C*), thus explaining the normal ERG amplitude observed. This finding implies that although *rdgB*$^{dgRNA}$ worked efficiently to generate germline deletions, it was not effective in tissue-specific depletion of RDGB protein levels. The reasons for this are not entirely clear. We speculate that RDGB is a protein with a long half-life and therefore despite efficient targeting of *rdgB* locus by the dgRNA in the eyes, the protein levels are not affected. However, it is also possible that the deletion of *rdgB* was not homogeneous in all eye precursor cells where Cas9 was expressed using *ey-Gal4*; residual photoreceptors with normal *rdgB* might be sufficient to generate a normal light response.

*skittles* (*sktl*) is a PI4P 5-kinase known to have a role during larval development (*Hassan et al., 1998*). Mutations in *sktl* are organismal lethal as well as cell lethal. Hence, we expected a strong phenotype when the *sktl* locus was deleted in eye discs using *ey-Gal4*. However, when sgRNA or dgRNA targeting *sktl* was used, we did not see any morphological phenotype in the eye [*Figure 4B* (iii)]. One possible reason for this could be that the *sktl* gRNAs was non-functional in eye discs, although they had worked well, being able to delete *sktl* in S2R$^+$ cells. In order to test this, we crossed *sktl* $^{sgRNA}$ to *nanos-Cas9* flies. From this cross, heterozygous F2 progeny were collected and their *skittles* genomic locus sequenced. Out of the 141 F2 flies sequenced, 132 showed in-frame indels with an intact kinase domain, with the largest deletion being 24 bp. This suggests that the *sktl* $^{sgRNA}$ were highly efficient at targeting *sktl* but only cells with at least a partially functional protein managed to survive during eye development and therefore appeared morphologically normal.

## Discussion

Patterning tissue architecture during metazoan development or the systemic control of animal physiology are complex processes and typically involves the function of genes acting in both cell-autonomous and non-cell autonomous modes. Identifying novel genes regulating these processes and uncovering their mode of action is facilitated by the ability to inactivate genes in specific cell types, tissue domains or organs with spatial and temporal precision. Such controlled inactivation can be achieved through the use of gene manipulation systems expressed with spatial and temporal precision using the GAL4/UAS system (*Brand and Perrimon, 1993*) and this approach has been coupled previously with methods to deplete specific RNAs to study their role in development and physiology (*Reim et al., 2014*). In this study, we present the use of the GAL4/UAS module for gene inactivation by using the CRISPR/Cas9 gene editing system. By expressing Cas9 with spatial and temporal precision using the GAL4/UAS system we were able to selectively inactivate genes in early precursor cells of the eye imaginal disc and thus uncover functions of the PI signaling system in the growth and patterning of the *Drosophila* eye. Using this method, we uncovered the function of 30 PI signaling genes in eye development and nine genes previously not implicated in this process. Our study identified phenotypes for genes such as *Dp110* and *Pten* that are previously described using either classical mutant alleles or using RNAi mediated depletion. However, the tool kit presented here includes reagents for editing ca. 72 PI binding proteins. While these have been described by in vitro protein biochemistry studies or predicted from structural bioinformatics, their function in vivo and their role in cell and developmental biology remains to be explored. Indeed, this resource can be used in almost any tissue or cell type in *Drosophila* that can be chosen for analysis, perhaps on the basis of their expression pattern or other relevant criteria. With this mind, we tested the ability of the dgRNAs in various tissues and found that any given dgRNA was able to generate deletions in diverse tissues such as the eye, wing and the germ line. Other groups have also recognized the

potential of CRISPR-based mutagenesis to perform tissue specific screens in *Drosophila* and are generating gRNA transgenic fly libraries. While some studies have focussed on transcription factors, protein kinases, protein phosphatases and genes implicated in human pathologies (*Port et al., 2020*), others have embarked on generating large-scale libraries with no special focus on any particular group of genes (*Meltzer et al., 2019*). Our study, however, represents the first comprehensive resource of editing tools that will allow a systems level analysis of PI signaling, a pathway of fundamental importance for cellular organization, tissue development, and architecture.

The system we have developed includes a number of innovative features. In the study of cell-cell interactions in development, it is desirable and indeed essential to mark the cell types in which gene manipulation is being performed. To facilitate this, we have designed an *UAS-Cas9-T2A-eGFP* construct which provides many advantages: (i) All cells expressing Cas9 also express eGFP thus marking cells guaranteed to express Cas9. The expression of eGFP from this construct is a definite readout of Cas9 expression unlike the expression of eGFP from an independent UAS-eGFP transgene. (ii) Since Cas9 and eGFP are expressed as independent proteins, the possibility of a Cas9 fusion protein with reduced function is eliminated. (iii) The *UAS-Cas9-T2A-eGFP* presented here avoids the need for two independent genetic elements to express both Cas9 and eGFP thus reducing the complexity of strain construction. (iv) The use of Cas9 with an NLS ensures that the protein is nuclear localized maximizing the probability of gene editing activity. Thus, our system offers the ability to generate groups of gene-edited cells/tissue with high-efficiency that are guaranteed to be marked by eGFP expression.

The genetic strategy for CRISPR/Cas9 editing presented here enhances the number of genetic elements that can be brought together in a single fly and allow for CRISPR based reagents to be used in innovative ways. For example, in addition to the basic eye development screen presented here as an illustrative example, Cas9 can be expressed in a mosaic fashion under the control of the Gal4/UAS system using the CoinFLP method (*Bosch et al., 2015*). This would allow for the generation of CRISPR edited mosaic clones useful for studying cell-cell interactions including cell competition that are integral elements of tissue development. Indeed, a recent study has presented evidence of a role of some genes involved in PI signaling as regulators of cell competition during eye development (*Janardan et al., 2020*) and the expanded repertoire of editing reagents generated during this study will allow the mechanism by which PI signaling regulates cell competition to be analysed in greater detail.

Recent studies have used the approach of having two sgRNAs both targeting the coding region immediately downstream of the start codon with the expectation of maximizing the likelihood of a frameshift mutation to achieve loss of function (*Port et al., 2020*). By contrast, we have taken a slightly different approach of using two sgRNAs, one located near the start codon and the other near the stop codon thus generating a deletion of the full open-reading frame post-editing. Even if only the gRNA targeting the first exon is functional, indels can be introduced and lead to loss-of-function from a resulting frameshift mutation. However, the presence of second sgRNA targeting the last exon will ensure that the entire gene is deleted thus increasing the chances of generating loss-of-function mutants. Secondly, because our dgRNA containing transgenic flies target both the first and the last exon and result in complete deletion of the gene, the subsequent introduction of a copy of the gene with homologous arms (*Baena-Lopez et al., 2013*) would result in gene knock-ins and reconstitution of function. This would require a germ line knockout of the gene and the use of our reagent set to generate such a null allele is described in the study. The use of a suitably mutagenized copy of the wild-type gene would allow individual point mutations or domain deletions of a protein to be generated, for example allowing the analysis of the role of specific amino acid residues or domains in a specific cell biological question. Since such knock-in constructs would also be expressed from the same locus as the wild-type gene thereby avoiding artifacts associated with altered levels of expression when transgenes are expressed from alternate genomic loci.

During the course of this study, we identified a few limitations for the use of these reagents. For example, in the case of the PI4P 5-kinase *sktl*, eye-specific deletion resulted in morphologically normal looking eyes. The reason for this was most likely that cells lacking *sktl* were eliminated (due to the essential nature of the gene product) early in eye disc development and only those cells with indels that are multiples of three bases thus possibly producing a partially functional protein survived. This is however dependent on the nature of the gene rather than the design of the gRNA

constructs. For example, when sgRNA targeting the kinase domain of *sktl* was used in conjunction with *ey >UAS-Cas9-GFP*, flies were pupal lethal presumably because even 1–2 bp deletions within the kinase domain of *sktl* abolishes kinase activity and gene function. In the case of the PI transfer protein RDGB, we were able to detect the presence of full-length protein in flies when eye specific knockout was performed. The possibility that either of the gRNAs were inefficient or non-functional in tissues was ruled out as we were able to use the *rdgB^dgRNA* flies to generate a germline knockout of this gene that phenocopied existing classical alleles. Since *rdgB* is located on the X-chromosome the inability to lose the protein by tissue-specific editing is unlikely to be due to editing of only one copy of the gene. We therefore speculate that the residual protein in eye-specific editing of *rdgB* most likely results from the long half-life of the protein that was present from early development. The implications of this is that when used for performing a screen in a tissue-specific manner, the lack of a phenotype does not conclusively rule out the involvement of that gene. Apart from the two examples described here, it is possible that CRISPR based targeting of genes in a tissue-specific manner may lead to heterogeneous disruption of the target gene especially in polyploid larval tissues such as the salivary glands and fat body. Therefore, absence of phenotypes in these tissue-specific gene disruptions using dgRNA will not be conclusive and under these circumstances, generation of a germline knock out would be valuable.

In summary, we have generated a toolkit consisting of a transgenic library of 103 dgRNA and a *UAS-Cas9-T2A-eGFP* that can be used in conjunction with the existing large repertoire of GAL4 lines to perform a systems level analysis of PI signaling selectively in any cell type/tissue of choice. The *UAS-Cas9-T2A-eGFP* construct would be helpful to track Cas9 expressing cells in culture or in fly tissues. The *dgRNAs* can also be transfected to delete gene function in cultured *Drosophila* cells. They can also be used to generate whole body knockouts and the null alleles so generated can be used as a template to knock in specific versions of the gene at the endogenous locus. While the PI kinases and phosphatases have been the subject of extensive analysis, the effectors of PI signaling, namely the binding proteins remain poorly studied. We envisage the availability of this dgRNA reagent set will accelerate the studies of the function of these proteins in cellular organization and tissue architecture. They will also facilitate the functional analysis of disease mechanisms in the case of those proteins linked to human disease.

# Materials and methods

## Key resources table

| Reagent type (species) or resource | Designation | Source or reference | Identifiers | Additional information |
|---|---|---|---|---|
| Genetic reagent (*D. melanogaster*) | *w^1118* | Bloomington *Drosophila* Stock Center | BDSC 3605 | |
| Genetic reagent (*D. melanogaster*) | *AB1-Gal4* | Bloomington *Drosophila* Stock Center | BDSC 1824 | *y^1 w*; P{GawB}AB1* |
| Genetic reagent (*D. melanogaster*) | *UAS-Cas9* | Bloomington *Drosophila* Stock Center | BDSC 54592 | *P{hsFLP}^1, y^1 w^1118; P{UAS-Cas9.P}attP2/TM6B, Tb^1* |
| Genetic reagent (*D. melanogaster*) | *attP40* | Bloomington *Drosophila* Stock Center | BDSC 25709 | *y^1 v^1 P{nos-phiC31\int.NLS}X; P{CaryP}attP40* |
| Genetic reagent (*D. melanogaster*) | *attP2* | Bloomington *Drosophila* Stock Center | BDSC 25710 | *P{nos-phiC31\int.NLS}X, y^1 sc^1 v^1 sev^21; P{CaryP}attP2* |
| Genetic reagent (*D. melanogaster*) | *ey-Gal4* | Bloomington *Drosophila* Stock Center | BDSC 5534 | *w*; P{GAL4-ey.H}3–8* |
| Genetic reagent (*D. melanogaster*) | *rdgB^9* | Bloomington *Drosophila* Stock Center | BDSC 27337 | |
| Genetic reagent (*D. melanogaster*) | *nos-Cas9* | Bloomington *Drosophila* Stock Center | BDSC 54591 | *y^1 M{nos-Cas9.P}ZH-2A w** |
| Genetic reagent (*D. melanogaster*) | *vasa-Cas9 (3)* | Bloomington *Drosophila* Stock Center | BDSC 51324 | *w^1118; PBac{vas-Cas9}VK00027* |
| Genetic reagent (*D. melanogaster*) | *vasa-Cas9 (1)* | Bloomington *Drosophila* Stock Center | BDSC 51323 | *y^1 M{vas-Cas9}ZH-2A w1118/FM7c* |

*Continued on next page*

*Continued*

| Reagent type (species) or resource | Designation | Source or reference | Identifiers | Additional information |
|---|---|---|---|---|
| Genetic reagent (*D. melanogaster*) | *Act5c-Cas9* | Bloomington *Drosophila* Stock Center | BDSC 54590 | $y^1$ M{Act5C-Cas9.P}ZH-2A w* |
| Genetic reagent (*D. melanogaster*) | *nub-Gal4;UAS-Cas9* | Bloomington *Drosophila* Stock Center | BDSC 67086 | w*; P{GawB}nubbin-AC-62; P{UAS-Cas9.P2}attP2 |
| Genetic reagent (*D. melanogaster*) | PI3K RNAi #1 | Bloomington *Drosophila* Stock Center | BDSC 61182 | $y^1$ sc* $v^1$ $sev^{21}$; P{TRiP.HMC05152}attP40 |
| Genetic reagent (*D. melanogaster*) | PI3K RNAi #2 | Bloomington *Drosophila* Stock Center | BDSC 27690 | $y^1$ $v^1$; P{TRiP.JF02770}attP2/TM3, $Sb^1$ |
| Cell line (*D. melanogaster*) | S2 R$^+$ | Drosophila *Genomics* Resource Center | Stock number 150 | |
| Antibody | Anti-Cas9 (Mouse monoclonal) | Takara | Cat#632628 (CloneTG8C1) | IF(1:200), WB (1:2000) |
| Antibody | Anti-OCRL (Rabbit polyclonal) | Gift from Avital Rodal | | IF(1:50) |
| Antibody | Anti-GFP (Rabbit polyclonal) | Abcam | Cat#: ab13970 | IF (1:5000) |
| Antibody | Anti-rdgB (Rat polyclonal) | *Yadav et al., 2015* | | WB (1:4000) |
| Antibody | Anti-GFP (Mouse monoclonal) | Santacruz | Cat # SC 9996 | WB (1:2000) |
| Antibody | Anti-Syntaxin A1 (Mouse monoclonal) | DHSB | Cat # 8C3 | WB (1:1000) |
| Antibody | Anti-beta tubulin | DHSB | Cat # E7C | WB (1:4000) |
| Recombinant DNA reagent | pBS-hsp70-Cas9 (plasmid) | Addgene | Cat# 46294 | |
| Recombinant DNA reagent | pAC-Cas9-sgRNA (plasmid) | Addgene | Cat# 49330 | |
| Recombinant DNA reagent | pBFvU6.2 (plasmid) | NIG (Japan) | | https://shigen.nig.ac.jp/fly/nigfly/cas9PlasmidsListAction.do |
| Recombinant DNA reagent | pBFvU6.2B (plasmid) | NIG (Japan) | | https://shigen.nig.ac.jp/fly/nigfly/cas9PlasmidsListAction.do |
| Recombinant DNA reagent | pUASt (plasmid) | *Brand and Perrimon, 1993* | | |
| Commercial assay or kit | Whole Genome DNA library prep kit | Illumina | FC-121–4002 | |
| Commercial assay or kit | Agilent high sensitivity DNA kit | Agilent | 5067–4626 | |
| Commercial assay or kit | Hiseq 2500 | Illumina | SY-401–2501 | |
| Software, algorithm | gRNA target finder | http://targetfinder.flycrispr.neuro.brown.edu/ | | |
| Software, algorithm | gRNA efficiency tool | https://www.flyrnai.org/evaluateCrispr/ | | |
| Other | DAPI stain | Invitrogen | D1306 | (1 µg/mL) |

### List of primers used in this study included as *Supplementary file 2*

gRNA design, synthesis, and transgenic generation gRNAs were cloned in pBFv6.2 and pBFv6.2B vectors (*Kondo and Ueda, 2013*), such that the gRNAs close to the start site were cloned in pBFv6.2 vector and the gRNAs close to the stop site were cloned in pBFv6.2B vector. A high-throughput method of gRNA cloning was devised, in which up to 5 gRNA constructs were cloned simultaneously in a single tube and then subsequently screened by PCR and sequencing. To generate the gRNA construct, the cloning vector was amplified using 5'-G(N19) GTTTTAGAGCTAGAAA TAGC-3' and 5'-GAAGTATTGAGGAAAACATA-3' oligos in order to introduce the 19 base CRISPR

target sequence into the vector. N19 corresponds to the sequences in gRNA target columns in *Supplementary file 1*. Once amplified, the DNA was purified, Dpn1 digested, phosphorylated at the ends using PNK, ligated and transformed. The bacterial colonies were screened using antisense primers for gRNA sequence (Complimentary to N19 sequence) and M13-rev primers by PCR. These were then sequence confirmed and tested in pairs for their efficacy in S2R$^+$ cells (see below). Efficient gRNA pairs were subcloned together as before (*Kondo and Ueda, 2013*) using EcoR1 and Not1 double digestion to generate dual-gRNA. Since the gRNAs were generated using PCR-based methods, which is intrinsically error prone, we sequenced the *attP* and vermillion region in addition to the gRNA region before generating transgenic flies. Transgenic flies expressing dual gRNA were generated using phiC31 integrase mediated transgenesis in either *attP40* (BDSC 25709) or *attP2* (BDSC 25710) flies (*Supplementary file 1*). The chromosome for transgenesis was chosen based on the chromosomal location of the gene. We tried not to use the same chromosome for the transgene where the gene is located.

S2R$^+$ cells transfection gRNA pairs were tested for the potential to delete the gene of interest in S2R$^+$ cells. Low passage S2R$^+$ cells were transfected with a mixture of 2 gRNA plasmids (100 ng/µl each) and hsp70-Cas9 (250 ng/µl) plasmids using Fugene HD transfection reagent according to the manufacturer's protocol. The transfected cells were incubated at room temperature for 48 hr and then collected by spinning in a tube and removing the medium. These were then resuspended in squishing buffer (10 mM Tris-Cl pH 8.2, 1 mM EDTA, 25 mM NaCl and 200 µg/ml Proteinase K) and incubated at 37°C for 1 hr and 95°C for 20 min. This was directly used for PCR to test the efficiency of deletion. Since the efficiency of transfection was found to be low, for most of the gRNAs, deletion bands were seen only when a nested PCR was performed. For the primer list used for nested PCRs, please refer to the key resource table. The deletion bands were sequenced and verified for deletion against reference genomic sequence.

Generation of *UAS-Cas9-T2A-eGFP* transgenes pUAST-eGFP vector was first generated by inserting eGFP sequence using EcoR1 and Xba1 restriction enzymes into the pUAST vector (*Brand and Perrimon, 1993*). Cas9 insert was PCR amplified from pAC-Cas9-sgRNA (*Bassett et al., 2013*) as a template, using the following primers (5'-gagcggagagcattgcggctgataagg-3' and 5'-gaggcgcaccgtgggccgcggccgcagatctctaccgctgccgctaccgctagca-3') which led to amplification of 3xFLAG-SV40-NLS-Cas9-NLS-T2A. The PCR product was digested using EcoR1 and Not1 enzymes and cloned into the pUAST-eGFP vector. After testing for its expression and efficacy in S2R$^+$ cells, transgenics were generated by P-element-based random insertion. Two lines were obtained and were both mapped to the third chromosome. One of these lines (with stronger mini white expression, as judged visually) was used for further experiments.

Isogenization of fly stocks for NGS *attP40* and *attP2* fly stocks were isogenized by crossing them to triple balancer fly stocks. *attP40* fly stocks (BDSC 25709) were triple balanced with *FM7a; CyO;Tb*. Subsequently, the balancer chromosomes were removed and stocks were maintained. After repeated attempts, we are unable to bring *attP2* stocks (BDSC 25710) on triple balancers and hence were unable to isogenize them. Hence, only *attP40* and S2R$^+$ cells were processed for NGS sequencing. *attP2* stocks were Sanger sequenced for genomic regions of interest whenever needed.

## Whole genome DNA library preparation, next-generation sequencing (NGS), and variant analysis

Genomic DNA was extracted using Qiagen Genomic DNA extraction kit. The Whole Genome DNA libraries were constructed using TruSeq Nano DNA LT Sample Preparation Kit Set B (24 Samples), Catalog no-FC-121–4002 according to the manufacturer's instructions (Illumina Inc, USA). Briefly, the genomic DNA (100 ng) were sonicated using Covaris S220 to 350 bp insert size and clean up the fragmented DNA using illumina sample purification beads. Purified fragmented DNA were end repaired, 3'ends adenylation and adapter ligation. DNA fragments were enriched using eight cycles of PCR according to the manufacturer's instructions. Library quality was analyzed on Agilent 2100 Bioanalyzer using DNA high sensitivity kit (Agilent Technologies, USA). Next-generation sequencing of libraries was performed using Illumina Hiseq 2500 for 2 × 100 bp. Sequence reads of both isogenized *attP40* (30 million paired end reads) and S2R$^+$ (29 million paired end reads) cells were imported into the CLC Genomics Workbench 12.0 (Qiagen). The reads were trimmed off the adapter sequences and reads containing low-quality bases eliminated. The trimmed raw sequences were mapped to

the *Drosophila melanogaster* genome and basic variant detection tools were used with default parameters of CLC Genomics workbench.

## Electroretinograms

Electroretinograms were performed as before (*Balakrishnan et al., 2018*). Briefly, flies were anesthetized and immobilized at the end of a disposable pipette tip using a drop of colorless nail varnish. Recordings were performed using glass microelectrodes (640786, Harvard Apparatus, MA) filled with 0.8% w/v NaCl solution. Voltage changes were recorded between the surface of the eye and an electrode placed on the thorax. At 1 day, flies, both male and female, were used for recording. Following fixing and positioning, flies were dark adapted for 5 min. ERGs were recorded with 2 s flashes of green light stimulus, with 10 stimuli (flashes) per recording and 15 s of recovery time between two flashes of light. Stimulating light was delivered from an LED light source to within 5 mm of the fly's eye through a fiber optic guide. Calibrated neutral density filters were used to vary the intensity of the light source. Voltage changes were amplified using a DAM50 amplifier (SYS-DAM50, WPI, FL) and recorded using pCLAMP 10.4. Analysis of traces was performed using Clampfit 10.4 (Molecular Devices, CA).

## Imaging of salivary glands and nephrocytes

Salivary glands or nephrocytes (of appropriate genotypes) from wandering 3rd-instar larvae were dissected in 1X phosphate-buffered saline (PBS) (pH 7.4) and fixed for 20 min at room temperature in 4% paraformaldehyde. Fixed tissues were then washed three times for 5 min in PBS (pH 7.4) and permeabilized in PBST (PBS + 0.3% Triton X-100) 3 × 5 min. Samples were blocked in 5%NGS in 0.3% PBST for 2 hr. Primary antibody incubation was performed overnight at 4°C in PBST with 5% normal goat serum. Primary antibody used are anti-dCas9 (Takara 632628) 1:200, anti-OCRL [Gift from Avital Rodal-(*Del Signore et al., 2017*)] 1:50, anti-eGFP (Abcam ab13970) 1:5000. Following washes in 0.3% PBST, incubation in secondary antibody for 2 hr in 0.3% PBST. Secondary antibody used is Alexa 633 Goat α-Rabbit IgG (Life Technology A21070) 1:200, Alexa 488 Goat α-chicken IgG (Abcam ab150173) 1:200, Alexa 568 Goat α-mouse IgG (Life Technology A11004). Following this the tissue was stained with DAPI for 5 min and mounted on a glass slide with 70% glycerol in PBS, sealed it with cover slip using colorless nail varnish, and allowed to dry. Samples were imaged on an Olympus FV3000 laser scanning confocal microscope under 40X/60 × 1.4 NA objective.

## Western blotting

Forty-eight hours post-transfection, S2R$^+$ cells were harvested by repeated flushing of the media and pelleted down in a 1.7 ml Eppendorf vial by centrifugation at a speed of 800 rpm for 10 min. The pelleted cells were lysed by repeated pipetting in Laemli loading buffer. Following this, the samples were heated at 95°C for 5 min and loaded onto an SDS-PAGE. The proteins were then transferred onto a nitrocellulose membrane and incubated overnight at 4°C with indicated antibodies. Primary antibody concentrations used were anti-dCas9 (Takara 632628) 1:2000; anti-GFP (Santa Cruz sc9996) 1:2000; anti-RDGB (*Yadav et al., 2015*), 1:4000; anti-Syntaxin A1 (DHSB 8C3) 1:1000 and anti-beta-Tubulin (DHSB E7c) 1:4000. The blots were then washed thrice with Tris Buffer Saline containing 0.1% Tween-20 (0.1% TBS-T) and incubated with appropriate HRP-conjugated secondary antibodies (Jackson Laboratories, Inc) at 1:10,000 for 45 min. After three washes with 0.1% TBS-T, blots were developed using Clarity Western ECL substrate on a GE ImageQuant LAS 4000 system.

## Generation of full body KO

Five Cas9 virgin flies (*Act5c-Cas9/vasa-Cas9/nos-Cas9*- see *Supplementary file 1*) were crossed to males of gRNA transgenic lines. ~ 10 F1 progeny males were crossed to balancer (specific for the gene of interest) flies. Subsequently, they were tested for deletion by nested PCR. F2 progeny from positive F1s were crossed to balancers (10 F2 flies were crossed from every F1 that showed a deletion band. Typically, not more than 50 crosses were set in case of many F1s being positive) and tested for deletion by PCR. Exact values for number of positive F2s per crosses set is given in

*Supplementary file 1*. Positives were tested for presence of gRNA and Cas9 transgenes and flies that lacked both were further developed into stocks.

## Nested PCR

Nested PCR was used to test the presence of deletion in S2R$^+$ cells and mosaic F1 flies. Single flies or S2R$^+$ cells were squished in a squishing buffer (see above), incubated at 37°C for 30 min and 95°C for 10 min. One μl of the supernatant was used to perform PCR with an outer set of primers (*Supplementary file 2*). One μl of this PCR product was used to perform PCR with an inner set of primers.

## Imaging *Drosophila* wings

Adult female flies of appropriate genotypes were dipped in 70% ethanol, briefly dried and wings clipped. The wings were mounted in colorless nail varnish and imaged under a light microscope (Olympus SZX12).

## Acknowledgements

This work was supported by the Department of Biotechnology, Government of India (BT/PRJ3748/GET/l 19/27/2015), a Wellcome-DBT India Alliance Senior Fellowship (IA/S/14/2/501540) to PR and the Department of Atomic Energy, Government of India, under project no. 12 R-D-TFR-5.04–08002 and 12 R-D-TFR-5.04–0900. We thank the *Drosophila*, Genomics and Imaging Core Facilities for extensive support in implementing this project. We thank Anitha VA and Srividhya A from the *Drosophila* facility for their help with completing this project.

## Additional information

### Funding

| Funder | Grant reference number | Author |
| --- | --- | --- |
| Department of Biotechnology , Ministry of Science and Technology | BT/PRJ3748/GET/l 19/27/2015 | Deepti Trivedi<br>Vinitha CM<br>Karishma Bisht<br>Vishnu Janardan<br>Padinjat Raghu |
| Wellcome Trust DBt India Alliance | IA/S/14/2/501540 | Vinitha CM<br>Karishma Bisht<br>Vishnu Janardan<br>Bishal Basak<br>Padinjat Raghu |
| National Centre for Biological Sciences | core | Deepti Trivedi<br>Vinitha CM<br>Karishma Bisht<br>Vishnu Janardan<br>Awadhesh Pandit<br>Bishal Basak<br>Padinjat Raghu |
| Department of Atomic Energy, Government of India | 12R-D-TFR-5.04–08002 | Padinjat Raghu |
| Department of Atomic Energy, Government of India | 12R-D-TFR-5.04–0900 | Padinjat Raghu |

The funders had no role in study design, data collection and interpretation, or the decision to submit the work for publication.

### Author contributions

Deepti Trivedi, Data curation, Formal analysis, Validation, Investigation, Visualization, Methodology, Writing - original draft, Project administration, Writing - review and editing; Vinitha CM, Karishma Bisht, Navyashree Ramesh, Investigation, Methodology; Vishnu Janardan, Data curation,

Investigation, Visualization, Methodology, Writing - original draft, Writing - review and editing; Awadhesh Pandit, Data curation, Investigation, Methodology; Bishal Basak, Investigation, Methodology, Writing - review and editing; Shwetha H, Investigation; Padinjat Raghu, Conceptualization, Supervision, Funding acquisition, Writing - original draft, Project administration, Writing - review and editing

### Author ORCIDs

Deepti Trivedi  https://orcid.org/0000-0001-8105-0728
Karishma Bisht  http://orcid.org/0000-0002-9088-1141
Padinjat Raghu  https://orcid.org/0000-0003-3578-6413

### Decision letter and Author response

Decision letter https://doi.org/10.7554/eLife.55793.sa1
Author response https://doi.org/10.7554/eLife.55793.sa2

## Additional files

### Supplementary files

• Supplementary file 1. List of all 103 *Drosophila* PI-signaling genes against which dgRNAs have been generated. The table indicates the CG numbers, gene names and what chromosome each of the genes are located on. For the phosphoinositide binding proteins, the various phosphoinositides either established or predicted to bind each protein have been listed. The table also has the closest human orthologs and associated human diseases from the Online Mendelian Inheritance in Man (OMIM) database indicated. The sequences of gRNA one and gRNA two for each gene are listed along with the size of the genomic deletion expected from these gRNA combinations, and the actual size of genomic deletion obtained in flies when crossed to *Act5c-Cas9*. The phenotypes obtained when these genes were deleted specifically in the eye and the efficiency (in terms of absolute numbers of flies with deletion as compared to total number of F2 flies) of whole fly gene knockout generation using either *Vasa-Cas9*, *Act5c-Cas9* or *Nos-Cas9* has been mentioned.

• Supplementary file 2. List of all oligonucleotides used in this study is provided. The sequence of each oligonucleotide is provided.

• Supplementary file 3. Isogenized *attP40* fly stock genomic sequence variants comparison with the reference genome. The table shows chromosome location wise comparison between the isogenized *attP40* stock with the reference genome. Average coverage for each sequence suggests the number of times each region has been sequenced. The frequency at which the variation is observed has been mentioned. A sequence quality of >Q30 score, which corresponds to >99.9% accuracy has been obtained. Each variation has been mapped based on genomic annotation of coding vs non-coding region. If variation in the coding region leads to amino-acid changes, it has been tabulated.

• Supplementary file 4. S2R$^+$ cell lines genomic sequence variants comparison with the reference genome. The table shows chromosome location wise comparison between the S2R$^+$ cells with the reference genome. Average coverage for each sequence suggests the number of times each region has been sequenced. The frequency at which the variation is observed has been mentioned. A sequence quality of >Q30 score, which corresponds to >99.9% accuracy has been obtained. Each variation has been mapped based on genomic annotation of coding vs non-coding region. If variation in the coding region leads to amino-acid changes, it has been tabulated.

• Transparent reporting form

### Data availability

Full genome sequencing for isogenized Attp40 Stock available at NCBI (BioProject ID PRJNA606147). Full genome sequencing for S2R+ cells available at NCBI (Bioproject ID PRJNA606149). Images for PI signaling genetic screen saved at Open Source Frame https://osf.io/pt7zu/.

The following datasets were generated:

| Author(s) | Year | Dataset title | Dataset URL | Database and Identifier |
|---|---|---|---|---|
| Trivedi D, Vinitha CM, Bisht K, Janardan V, Pandit A, Basak B, Raghu P | 2020 | Full genome sequencing for isogenized Attp40 Stock | https://www.ncbi.nlm.nih.gov/bioproject/PRJNA606147/ | NCBI BioProject, BioProjectIDPRJNA606147 |
| Trivedi D, Vinitha CM, Bisht K, Janardan V, Pandit A, Basak B, Raghu P | 2020 | Full genome sequencing for S2R+ cells | https://www.ncbi.nlm.nih.gov/bioproject/PRJNA606149/ | NCBI BioProject, BioProjectIDPRJNA606149 |
| Trivedi D, Bisht K, Janardan V, Pandit A, Basak B, Raghu P, Vinitha CM | 2020 | A genome engineering resource to uncover principles of cellular organization and tissue architecture by lipid signalling | https://osf.io/pt7zu/ | Open Science Framework, pt7zu |

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
