## [Decision Letter]

**Acceptance summary:**

Phosphoinositide (PI) signaling is important for various biological processes in all metazoan organisms. The authors generated a CRISPR-based genome engineering toolkit to make tissue-specific mutations in π signaling genes in *Drosophila*. The reagents generated in this study will be a useful and versatile genetic tool for researchers in the field.

**Decision letter after peer review:**

Thank you for submitting your article "A genome engineering resource to uncover principles of cellular organization and tissue architecture by lipid signalling" for consideration by *eLife*. Your article has been reviewed by three peer reviewers, one of whom is a member of our Board of Reviewing Editors, and the evaluation has been overseen by Utpal Banerjee as the Senior Editor. The reviewers have opted to remain anonymous.

The reviewers have discussed the reviews with one another and the Reviewing Editor has drafted this decision to help you prepare a revised submission.

Three reviewers suggested major criticisms on validations of the new tool and generated stocks. Therefore, the revised manuscript should demonstrate efficient knockout events by expression analysis and by extensive molecular analysis of targeted mutations for many more genes rather than a few. Essential revisions are summarized as below and full reviewers' comments are attached for your reference.

Summary:

This study describes the generation of a CRISPR/Cas-9 mediated gene editing toolkit in *Drosophila* and offers germline or tissue-specific knockouts of *Drosophila* π signaling genes. There have been several independent efforts to generate systematic knockout tools in this study, and it will be a valuable toolkit for the fly community working on π signaling in *Drosophila* cells and whole animals. However, this report shows a few successful cases of gene knockouts and insufficient assessments for the new tool, and therefore, the manuscript requires additional extensive molecular and genetic evaluation of the presented data.

Essential revisions:

1) Confirm DNA lesions in many more cases:

i) The authors designed dgRNA constructs to generate gene deletion. Although they claimed that the targeted deletions were confirmed in S2R+ cells, it is unclear whether the mutations were confirmed in flies.

According to subsection “Investigating the role of π signalling in eye development using the dgRNA toolkit”, a few representative ey>Cas9;dgRNA lines were tested for genomic deletion. Then, it appears that the majority of 103 transgenic lines were not tested for molecular lesions. Table 1 shows a list of genes with the size of the genomic deletion. If in fact only a few lines were tested, then it would be difficult to properly evaluate the data in Table 1. It is important to know how many lines were tested for deletion and how many actually showed an expected size of deletions.

2) Better validation of the efficacy:

i) The authors verified the efficiency and usefulness of the tool by targeting π genes in the eye. Most of the known target genes resulted in desirable phenotypes in the eye; however, some did not. In this case, it could be a matter of protein stability as the authors explained, but it also could be due to the efficiency or the variability of the new tool in the eye. The tool requires a better validation on a cell-by-cell basis. For example, with the use of in situ hybridization or an antibody against the protein (there are several antibodies readily available), expressions of target genes could be measured. If the phenotype/expression level could be quantified, efficiencies of whole body mutant, tissue-specific RNAi, and tissue-specific deletion need to be compared.

ii) In Figure 4A, the authors have shown a truncated band of CG5734 without showing its full-length gene. Even with the presence of truncated size, the intact gene may be still viable in some eye cells. It again will be important to show how efficient it is to use the new tool at a tissue level.

iii) In addition to ey-gal4, the tool requires validations with additional tissue-specific drivers. This will not only verify its effectiveness but enhance the broader applications.

iv) In subsection “Generation of whole fly knockouts and using dgRNA transgenics”, it seems that they tried germline knockout for several genes, but the procedure is not mentioned and the data are described very ambiguously. "The deletions obtained in this manner are highly efficient (between 4-100% efficiency in F1 flies and between 3-70% efficiency in F2 flies) in generating germline knockouts with deletions ranging in size between 400bp and 38kb." From this description, it is difficult to know whether the deletions are highly efficient as claimed. The procedure for making germline knockout and the results need to be described in full detail. In Table 1, a number of tissue-specific knockouts show no phenotype. It is possible that some of them may not have expected mutations.

v) There is no data showing the loss of protein expression in eye discs from eye-specific knockout lines. Crosses with ey-Gal4 will generate mosaic tissues in developing eye discs. It is important to know how efficient is a somatic knockout in the region of ey-Gal4 expression.

3) Compare SNPs in other strains:

i) The authors make the argument that because their dgRNA constructs work in both their BL25709 *Drosophila* and S2R+ cells and match the reference genome, that they should work in other genetic backgrounds. However, these backgrounds don't sample the full range of nucleotide variation in laboratory stocks. Can the authors compare their gRNA sequences with genomes of other common lab strains (or optionally other wildtype genomic sequences like the DGRP)?

ii) Authors have carefully selected dgRNA sequences by comparing genomic sequences of S2R+ cells, parent fly stocks, and reference fly genome. This will be a useful resource for researchers who would refer to this study in the future. Therefore, detailed descriptions of SNPs found (e.g. in a table or additional diagrams) need to be added in addition to the diagram shown in Figure 2.

4) Separate methods from results:

i) Since the manuscript describes methods in the result, it does not separate the methods from the main text. However, it should either be more specific in the text or include the methods part for further details. As one example, UAS-Cas9-eGFP transgenic flies newly generated in this study require detail information for those who will order stocks.

Reviewer #1:

In this manuscript, Gaghu and colleagues have designed a set of CRISPR based genome editing tools and constructed reagents by which 103 phosphoinositide (PI)-related genes can be individually deleted. With the use of the reagents, authors have identified novel π genes that control eye development and verified the effectiveness of the tools.

This study will potentially benefit the fly community by providing the new tissue-specific CRISPR tool. However, as a resource paper, the manuscript requires better validations on the technique itself and focus on describing the validity of the tool.

1) Authors have carefully selected dgRNA sequences by comparing genomic sequences of S2R+ cells, parent fly stocks, and reference fly genome. This will be a useful resource for researchers who would refer to this study in the future. Therefore, detailed descriptions of SNPs found (e.g. in a table or additional diagrams) need to be added in addition to the diagram shown in Figure 2.

a) In the same vein, if SNPs in different backgrounds are critical for the CRISPR efficiency, giving a few examples of such cases would enhance the importance of resources provided.

2) Since the manuscript describes methods in the result, it does not separate the methods from the main text. However, it should either be more specific in the text or include the methods part fur further details. As one example, UAS-Cas9-eGFP transgenic flies newly generated in this study require detail information for those who will order stocks.

3) The authors verified the efficiency and usefulness of the tool by targeting π genes in the eye. Most of the known target genes resulted in desirable phenotypes in the eye; however, some did not. In this case, it could be a matter of protein stability as the authors explained, but it also could be due to the efficiency or the variability of the new tool in the eye. The tool requires a better validation in a cell-by-cell basis. For example, with the use of in situ hybridization or an antibody against the protein, expressions of target genes could be measured. If the phenotype/expression level could be quantified, efficiencies of whole body mutant, tissue-specific RNAi, and tissue-specific deletion need to be compared.

a) In Figure 4A, the authors have shown a truncated band of CG5734 without showing its full-length gene. Even with the presence of truncated size, the intact gene may be still viable in some eye cells. It again will be important to show how efficient it is to use the new tool at a tissue level.

4) In addition to ey-gal4, the tool requires validations with additional tissue-specific drivers. This will not only verify its effectiveness but enhance the broader applications.

Reviewer #2:

This study offers a library of 103 gRNA lines to generate germline or tissue-specific knockouts of *Drosophila* π signaling genes. With additional constructs, they could also be used to generate new alleles or tagged proteins. This is a useful toolkit for the growing community working on signalling, transport and metabolism of π in *Drosophila* cells and whole animals. The lines target two sites in each gene, are robust to some target genome variation, and in some cases are validated using RNAi or known mutant phenotypes. This is a very useful collection of well-designed lines, with good supporting validation.

The authors make the argument that because their dgRNA constructs work in both their BL25709 *Drosophila* and S2R+ cells and match the reference genome, that they should work in other genetic backgrounds. However, these backgrounds don't sample the full range of nucleotide variation in laboratory stocks. Can the authors compare their gRNA sequences with genomes of other common lab strains (or optionally other wildtype genomic sequnces like the DGRP)?

Reviewer #3:

This manuscript describes the generation of a genetic tool kit for tissue-specific and whole animal knockout using CRISPR/Cas9-mediated gene editing in *Drosophila*. There have been several independent efforts to generate Cas9-based systematic gene knockout tools in culture cells and flies. This study attempts to demonstrate the feasibility of constructing guide RNAi library targeted to a relatively small set the genes involved in phosphoinositides (PI) metabolism and signaling. This report shows a few successful cases of gene knockout and a list of 103 targeted genes. Therefore, in principle, the genetic tools generated in this study can be valuable resources in the field of π signaling. However, I feel that the general strategy used in this study is not particularly novel, and more importantly, analysis of their library appears to be rather limited and preliminary for proper evaluation of the presented data.

Merits of this work:

1) They sequenced S2R+ genome and an isogenic fly genome. After comparing these sequences and the reference genome, gRNA targets were carefully selected to avoid potential mismatches or off-targets.

2) This study used double guide RNAs targeted to 1st and last coding exons to generate null mutations by deletion. The use of T2A-mediated eGFP expression also helps to identify Cas9-expressing cells.

3) Cas9 mutagenesis targeted to a full set of PI-related genes is a new attempt.

Weaknesses:

Overall, their strategy for tissue-specific knockout is similar to two recent papers (Meltzer et al., 2019; Port et al., 2020). Hence, novelty of this manuscript is somewhat diminished. In addition, there are major issues related to their methods and data, as listed below.

1) Authors designed dgRNA constructs to generate gene deletion. Although they claimed that the targeted deletions were confirmed in S2R+ cells, it is unclear whether the mutations were confirmed in flies.

According to subsection “Investigating the role of π signalling in eye development using the dgRNA toolkit”, a few representative ey>Cas9;dgRNA lines were tested for genomic deletion. Then, it appears that the majority of 103 transgenic lines were not tested for molecular lesions. Table 1 shows a list of genes with the size of the genomic deletion. If in fact only a few lines were tested, then it would be difficult to properly evaluate the data in Table 1. It is important to know how many lines were tested for deletion and how many actually showed expected size of deletions.

2) In subsection “Generation of whole fly knockouts and using dgRNA transgenics”, it seems that they tried germline knockout for several genes, but the procedure is not mentioned and the data are described very ambiguously. "The deletions obtained in this manner are highly efficient (between 4-100% efficiency in F1 flies and between 3-70% efficiency in F2 flies) in generating germline knockouts with deletions ranging in size between 400bp and 38kb." From this description, it is difficult to know whether the deletions are highly efficient as claimed. The procedure for making germline knockout and the results need to be described in full detail. In Table 1, a number of tissue-specific knockout show no phenotype. It is possible that some of them may not have expected mutations.

3) There is no data showing the loss of protein expression in eye discs from eye-specific knockout lines. Crosses with ey-Gal4 will generate mosaic tissues in developing eye discs. It is important to know how efficient is somatic knockout in the region of ey-Gal4 expression.

[Editors' note: further revisions were suggested prior to acceptance, as described below.]

Thank you for resubmitting your work entitled "A genome engineering resource to uncover principles of cellular organization and tissue architecture by lipid signalling" for further consideration by *eLife*. Your revised article has been evaluated by Utpal Banerjee (Senior Editor) and a Reviewing Editor.

The manuscript has been improved but there are some remaining issues that need to be addressed before acceptance, as outlined below:

All the reviewers agreed that the authors added a substantial amount of new data and appropriately addressed most of the concerns.

Reviewer #1:

In the revised manuscript, Trivedi et al. provide a significantly improved version of their work with additional analyses and in vivo validations and addressed most of the concerns raised by the reviewers. Two additional points need to be modified to clarify the results.

1) In Figure 2B, the authors validated the efficiency of ocrl deletion by immunohistochemistry and showed that ocrl knockout reduces the anti-OCRL staining in the wing disc. However, staining patterns shown in controls are too low, and it is hard to tell whether the antibody used was working in the disc. Providing a reference for the antibody that shows an expression pattern in a representative organ and/or highlighting higher mag images will be critical to reassure the data. If this antibody was first used in an IHC experiment, proper confirmations for the antibody would be needed.

2) In Figure 3C, gel images do not have proper labels.

3) If possible, it will be best to organize the deposition of generated flies to major stock centers before publication and describe it in the text.

Reviewer #2:

The authors have added substantial new validation, particularly on the areas of SNP variation, the lesions obtained, and homozygous phenotypes. This now gives the reader greater confidence in the usefulness of their resource – for somatic knockouts one can often find a localized phenotype but not always, perhaps due to perdurance of a stable gene product; and most of their gRNA stocks can be used to generate germline knockouts, and they have established mutant stocks for about half of them.

Taken together, I think this collection will be a valuable resource targeting many genes of great interest for the fly signaling and lipid communities. The paper now gives a much better sense of the approach's strengths and limitations, and there is enough information to make many lines of wide use immediately and to develop further mutant stocks for the remainder. To make these lines accessible to the community, I hope that both the gRNA and mutant stocks will be offered promptly to the major stock centers, if the authors haven't done so already.

Reviewer #3:

1) The authors have revised the manuscript with additional experiments and clarifications. The revised manuscript appears to be significantly improved by extensive molecular analysis of targeted mutations in fly tissues. Phenotype analysis with additional GAL4 drivers also supports the effectiveness of the CRISPR/Cas9 tools.

2) Considering that the majority of targeted genes by ey-GAL4 do not show phenotypes, it is still possible that targeting might not have caused sufficient loss of protein expression in many genes. In this regard, checking the expression level for only one gene (Ocrl) is a weak point. However, the new data showing deletions in most targeted genes seems to compensate for the weakness. In summary, the revised manuscript seems to be acceptable, but a few mistakes listed below should be corrected.

3) Some genotypes in figures are not italicized. Those genotypes need to be fixed.

4) Anti-Ocrl staining appears to be nuclear staining to me, although it is too weak to be certain. A better way to show the mutation effect is to use a different Gal4 such as en-Gal4 to show an internal control. It would be good to show a test for antibody specificity.

---

## [Author Response]

Essential revisions:1) Confirm DNA lesions in many more cases:i) The authors designed dgRNA constructs to generate gene deletion. Although they claimed that the targeted deletions were confirmed in S2R+ cells, it is unclear whether the mutations were confirmed in flies.According to subsection “Investigating the role of π signalling in eye development using the dgRNA toolkit”, a few representative ey>Cas9;dgRNA lines were tested for genomic deletion. Then, it appears that the majority of 103 transgenic lines were not tested for molecular lesions. Table 1 shows a list of genes with the size of the genomic deletion. If in fact only a few lines were tested, then it would be difficult to properly evaluate the data in Table 1. It is important to know how many lines were tested for deletion and how many actually showed an expected size of deletions.

To address the above comment following has been done:

a) dgRNA for all the 103 genes were tested for deletions in flies after crossing with Act5c-Cas9. 102 of these seem to show deletion in flies. Thus, in addition to our previous experiments (reported in the initial submission) where we showed deletions generated by all of the dgRNAs in S2R+ cells, we have now included new data showing that 102/103 dgRNAs also work in fly tissues (Supplementary file 1).

b) The exact size of deletion for each dgRNA is now added in Supplementary file 1 according to the results obtained in flies after sequencing the deletion.

2) Better validation of the efficacy:i) The authors verified the efficiency and usefulness of the tool by targeting π genes in the eye. Most of the known target genes resulted in desirable phenotypes in the eye; however, some did not. In this case, it could be a matter of protein stability as the authors explained, but it also could be due to the efficiency or the variability of the new tool in the eye. The tool requires a better validation on a cell-by-cell basis. For example, with the use of in situ hybridization or an antibody against the protein (there are several antibodies readily available), expressions of target genes could be measured. If the phenotype/expression level could be quantified, efficiencies of whole body mutant, tissue-specific RNAi, and tissue-specific deletion need to be compared.

We agree with this point. It has been a challenge to obtain appropriate antibodies during the current pandemic with many labs shut or difficult shipping conditions. Therefore, we were only able to perform a limited number of experiments. A well-validated antibody for OCRL was available. We were able to use this and show that in discs where the dgRNA for *ocrl* was expressed, there was a reduction in OCRL protein expression (Figure 2B). Quantification of the staining intensity across discs (Figure 2C) showed that the reduction in OCRL protein levels was quite uniform suggested minimal cell to cell variability in OCRL protein depletion with these reagents.

ii) In Figure 4A, the authors have shown a truncated band of CG5734 without showing its full-length gene. Even with the presence of truncated size, the intact gene may be still viable in some eye cells. It again will be important to show how efficient it is to use the new tool at a tissue level.

This has been tested for several genes. Most do not show wildtype bands either because of high efficiency of deletion, or more likely because smaller deletion products are amplified favorably in PCR cycles. However, we have added a Figure 4—figure supplement 1 with an example where the WT product was amplified.

iii) In addition to ey-gal4, the tool requires validations with additional tissue-specific drivers. This will not only verify its effectiveness but enhance the broader applications.

We also performed additional deletions with tissue specific GAL4 lines. We expressed three different guideRNAs for the PI3K signalling pathway using nub-GAL4 that expresses in a specific domain of the developing wing imaginal disc. Each of these dgRNAs (encoding components of the Class I PI3K signalling pathway), when expressed in the developing eye disc with UAS-Cas9 had resulted in changes in tissue growth resulting in altered eye size. Likewise, when expressed using nub-GAL4, they resulted in equivalent effects in the phenotype of the adult wing. Thus, deletion of *pTEN^dgRNA^* resulted in a larger wing blade while deletion of *pi3K92E^dgRNA^*and *akt^dgRNA^*resulted in substantial reductions in wing blade size. These results are now presented in Figure 4E.

Overall, during the course of this study, we have shown that our dgRNAs when expressed using four tissue specific drivers (ey-GAL4, nub-GAL, vasa-GAL4, nanos-GAL4) produce deletions and phenotypes. In addition, we have also demonstrated its ability to produce deletions with the ubiquitous Act5C-Gal4 that expresses in all tissues.

iv) In subsection “Generation of whole fly knockouts and using dgRNA transgenics”, it seems that they tried germline knockout for several genes, but the procedure is not mentioned and the data are described very ambiguously. "The deletions obtained in this manner are highly efficient (between 4-100% efficiency in F1 flies and between 3-70% efficiency in F2 flies) in generating germline knockouts with deletions ranging in size between 400bp and 38kb." From this description, it is difficult to know whether the deletions are highly efficient as claimed. The procedure for making germline knockout and the results need to be described in full detail.

The absolute numbers have now been added to Supplementary file 1.

In Table 1, a number of tissue-specific knockouts show no phenotype. It is possible that some of them may not have expected mutations.

We have a number of examples where there was no phenotype, but testing for deletions by PCR in these showed that a deletion had indeed taken place. This has now been added as Figure 4—figure supplement 2. It is possible that in some of these cases (i) the gene in question does not have a role in eye development (ii) some amount of residual pre-deletion RNA/protein perdures in the cells and is sufficient to support normal function.

v) There is no data showing the loss of protein expression in eye discs from eye-specific knockout lines. Crosses with ey-Gal4 will generate mosaic tissues in developing eye discs. It is important to know how efficient is a somatic knockout in the region of ey-Gal4 expression.

Due to the challenge of obtaining appropriate antibodies during the current pandemic with many labs shut or difficult shipping conditions, we were only able to perform a limited number of experiments. A well- validated antibody for OCRL was available. We were able to use this and show that in discs where the dgRNA for ocrl was expressed there was a reduction in OCRL protein expression (Figure 2B). Quantification of the staining intensity across discs (Figure 2C) showed that the reduction in OCRL protein levels was quite uniform. Please note that we were able to do this experiment with only 2 genotypes- Act5c-Cas9/+; *ocrl^dgRNA^*/+ and Act5c-Cas9/+ due to limitation of antibody availability.

3) Compare SNPs in other strains:i) The authors make the argument that because their dgRNA constructs work in both their BL25709 *Drosophila* and S2R+ cells and match the reference genome, that they should work in other genetic backgrounds. However, these backgrounds don't sample the full range of nucleotide variation in laboratory stocks. Can the authors compare their gRNA sequences with genomes of other common lab strains (or optionally other wildtype genomic sequences like the DGRP)?

We have compared the sequences obtained using NGS with WT genomic seq from Flybase. In addition, the designed gRNAs have now been matched to recently added genomic sequences of reference genome r_6, vasa-Cas9 (BDSC 51324), nos-Cas9 II (BDSC 78781) and nos-Cas9 III (BDSC 78782). Only 1 gRNA target (targeting CG10426), out of the 206 generated, seemed to have one base mismatch (Supplementary file 1- the variant base marked in red ), while the other 205 targets matched between all the genomes. These gRNA can hence be used in many genetic backgrounds.

ii) Authors have carefully selected dgRNA sequences by comparing genomic sequences of S2R+ cells, parent fly stocks, and reference fly genome. This will be a useful resource for researchers who would refer to this study in the future. Therefore, detailed descriptions of SNPs found (e.g. in a table or additional diagrams) need to be added in addition to the diagram shown in Figure 2.

The detailed description of all SNP’s found have been added as Figure 2; Supplementary file 3 and 4.

4) Separate methods from results:i) Since the manuscript describes methods in the result, it does not separate the methods from the main text. However, it should either be more specific in the text or include the methods part for further details. As one example, UAS-Cas9-eGFP transgenic flies newly generated in this study require detail information for those who will order stocks.

A separate section of Materials and methods has now been added to detail all the experimental protocols.

Reviewer #1:In this manuscript, Gaghu and colleagues have designed a set of CRISPR based genome editing tools and constructed reagents by which 103 phosphoinositide (PI)-related genes can be individually deleted. With the use of the reagents, authors have identified novel π genes that control eye development and verified the effectiveness of the tools.This study will potentially benefit the fly community by providing the new tissue-specific CRISPR tool. However, as a resource paper, the manuscript requires better validations on the technique itself and focus on describing the validity of the tool.1) Authors have carefully selected dgRNA sequences by comparing genomic sequences of S2R+ cells, parent fly stocks, and reference fly genome. This will be a useful resource for researchers who would refer to this study in the future. Therefore, detailed descriptions of SNPs found (e.g. in a table or additional diagrams) need to be added in addition to the diagram shown in Figure 2.

This has been done. Discussed in Essential Revisions.

a) In the same vein, if SNPs in different backgrounds are critical for the CRISPR efficiency, giving a few examples of such cases would enhance the importance of resources provided.

We found only one example of a guide that had a mismatch due to a strain specific SNP. This has been explained in the text.

2) Since the manuscript describes methods in the result, it does not separate the methods from the main text. However, it should either be more specific in the text or include the methods part fur further details. As one example, UAS-Cas9-eGFP transgenic flies newly generated in this study require detail information for those who will order stocks.

A separate Materials and methods section has been added.

3) The authors verified the efficiency and usefulness of the tool by targeting π genes in the eye. Most of the known target genes resulted in desirable phenotypes in the eye; however, some did not. In this case, it could be a matter of protein stability as the authors explained, but it also could be due to the efficiency or the variability of the new tool in the eye. The tool requires a better validation in a cell-by-cell basis. For example, with the use of in situ hybridization or an antibody against the protein, expressions of target genes could be measured. If the phenotype/expression level could be quantified, efficiencies of whole body mutant, tissue-specific RNAi, and tissue-specific deletion need to be compared.a) In Figure 4A, the authors have shown a truncated band of CG5734 without showing its full-length gene. Even with the presence of truncated size, the intact gene may be still viable in some eye cells. It again will be important to show how efficient it is to use the new tool at a tissue level.4) In addition to ey-gal4, the tool requires validations with additional tissue-specific drivers. This will not only verify its effectiveness but enhance the broader applications.Reviewer #3:This manuscript describes the generation of a genetic tool kit for tissue-specific and whole animal knockout using CRISPR/Cas9-mediated gene editing in *Drosophila.* There have been several independent efforts to generate Cas9-based systematic gene knockout tools in culture cells and flies. This study attempts to demonstrate the feasibility of constructing guide RNAi library targeted to a relatively small set the genes involved in phosphoinositides (PI) metabolism and signaling. This report shows a few successful cases of gene knockout and a list of 103 targeted genes. Therefore, in principle, the genetic tools generated in this study can be valuable resources in the field of π signaling. However, I feel that the general strategy used in this study is not particularly novel, and more importantly, analysis of their library appears to be rather limited and preliminary for proper evaluation of the presented data.Merits of this work:1) They sequenced S2R+ genome and an isogenic fly genome. After comparing these sequences and the reference genome, gRNA targets were carefully selected to avoid potential mismatches or off-targets.2) This study used double guide RNAs targeted to 1st and last coding exons to generate null mutations by deletion. The use of T2A-mediated eGFP expression also helps to identify Cas9-expressing cells.3) Cas9 mutagenesis targeted to a full set of PI-related genes is a new attempt.Weaknesses:Overall, their strategy for tissue-specific knockout is similar to two recent papers (Meltzer et al., 2019; Port et al., 2020). Hence, novelty of this manuscript is somewhat diminished. In addition, there are major issues related to their methods and data, as listed below.1) Authors designed dgRNA constructs to generate gene deletion. Although they claimed that the targeted deletions were confirmed in S2R+ cells, it is unclear whether the mutations were confirmed in flies.According to subsection “Investigating the role of π signalling in eye development using the dgRNA toolkit”, a few representative ey>Cas9;dgRNA lines were tested for genomic deletion. Then, it appears that the majority of 103 transgenic lines were not tested for molecular lesions. Table 1 shows a list of genes with the size of the genomic deletion. If in fact only a few lines were tested, then it would be difficult to properly evaluate the data in Table 1. It is important to know how many lines were tested for deletion and how many actually showed expected size of deletions.

This has been done and added to Supplementary file 1.

2) In subsection “Generation of whole fly knockouts and using dgRNA transgenics”, it seems that they tried germline knockout for several genes, but the procedure is not mentioned and the data are described very ambiguously. "The deletions obtained in this manner are highly efficient (between 4-100% efficiency in F1 flies and between 3-70% efficiency in F2 flies) in generating germline knockouts with deletions ranging in size between 400bp and 38kb." From this description, it is difficult to know whether the deletions are highly efficient as claimed. The procedure for making germline knockout and the results need to be described in full detail. In Table 1, a number of tissue-specific knockout show no phenotype. It is possible that some of them may not have expected mutations.

The process for making germ line knockouts has been described under Materials and methods Figure 4—figure supplement 2 now included showing examples where a gene gave no phenotype but PCR analysis clearly detected a deletion

3) There is no data showing the loss of protein expression in eye discs from eye-specific knockout lines. Crosses with ey-Gal4 will generate mosaic tissues in developing eye discs. It is important to know how efficient is somatic knockout in the region of ey-Gal4 expression.

We have now included protein expression data for an example-*ocrl* where inducing a deletion resulted in depletion of protein. Depletion has been quantified (Figure 2B and 2C).

[Editors' note: further revisions were suggested prior to acceptance, as described below.]

Reviewer #1:In the revised manuscript, Trivedi et al. provide a significantly improved version of their work with additional analyses and in vivo validations and addressed most of the concerns raised by the reviewers. Two additional points need to be modified to clarify the results.1) In Figure 2B, the authors validated the efficiency of ocrl deletion by immunohistochemistry and showed that ocrl knockout reduces the anti-OCRL staining in the wing disc. However, staining patterns shown in controls are too low, and it is hard to tell whether the antibody used was working in the disc. Providing a reference for the antibody that shows an expression pattern in a representative organ and/or highlighting higher mag images will be critical to reassure the data. If this antibody was first used in an IHC experiment, proper confirmations for the antibody would be needed.

Under the present circumstances, it has proved very, very difficult to locate suitable antibodies and procure them for these experiments. Sometimes, when antibodies are described in the literature and apparently readily available, labs are closed/not-accessible or there are shipping restrictions in place. This is a genuine and serious problem limiting our ability to carry out additional experiments. We request all reviewers and editors to please understand the situation.

We happened to have some of the ocrl antibody previously procured for another project and have used what was left for a single experiment. Each point on the graph is a single disc.

However, to address the scientific question being addressed and reassure reviewers that the staining being presented in Figure 2B.C is both specific and reproducible, we present data previously generated using these same reagents that should answer the questions being posed by the reviewers.

As part of a project on the renal phenotypes of ocrl, modelled in *Drosophila,* we have been working with both a germline knockout of ocrl as well as a nephrocyte specific knockout generated using these same reagents. carried out more detailed analysis using these reagents. This is part of another manuscript under process right now. We present it here to reassure the reviewer that the ocrl antibody and gRNA reagents do reproducibly result in protein depletion when used for tissue specific depletion and that the result shown in the paper with the disc. These data are part of a separate manuscript and presented here in Author response image 1.

**Author response image 1. sa2fig1:** Verification of OCRL protein depletion when using CRISPR/Cas9 genome engineering reagents. (A) Western blot from protein lysates of 3rd instar larvae from wild type and dOCRLKO. The dOCRL polypeptide of the expected Mr of ca. 100 kDa is completely absent in dOCRLKO lysates. Protein levels of actin are used as the loading control. dOCRL antibody used here has been described in (Del Signore et al., 2017)(B) Immunolabelling of pericardial nephrocytes from 3rd instar larvae to examine the distribution of the dOCRL protein. As expected, the dOCRL protein is seen in punctate structures throughout the cell body but excluded from the nucleus. This staining is completely absent in dOCRLKO animals. The experiment was repeated three times and multiple animals examined in each case. A single nephrocyte is shown for illustrative purposes. (C) Schematic depicting the generation of nephrocyte specific knock out (NephroctyeKO) of dOCRL. A nephrocyte specific Gal4-Dot-Gal4 is used and combined with the dgRNA for dOCRL and UAS-Cas9-eGFP. Animals of the test and control genotype are identified on the basis of eGFP expression. (D) PCR analysis demonstrating the 555 bp band detected when dOCRL is edited by these dgRNAs is shown. For nephrocyteKO animals larvae were dissected and the body wall (including heart tube and nephrocytes) was sued for the DNA prep. As a positive control the germ line dOCRLKO larval lysates were used. CG7004 is used as a control gene for the quality and quantity of the DNA prep. The 555bp diagnostic band is seen in the dOCRLKO and nephrocyteKO lysates but not in wild type flies and in NTC (no template control). (E) Immunostaining of nephrocytes in dOCRL-NephrocyteKO animals. A single nephrocyte is shown. Green channel shows the GFP generated from the Cas9-EGFP transgene. Red channel shows immunolabelling with the dOCRL antibody. Nucleus is stained with DAPI. There is no detectable signal for dOCRL in the nephrocytes of NephrocyteKO animals; experiment repeated twice and 24 individual nephrocytes images from multiple animals. Although the immunostaining for dOCRL has been repeated twice, NephrocyteKO animals have been generated many times for physiological and cell biological characterization which is described in that manuscript.

2) In Figure 3C, gel images do not have proper labels.

Labelling has now been done.

3) If possible, it will be best to organize the deposition of generated flies to major stock centers before publication and describe it in the text.

We are already in touch with the DGRC to deposit the gRNA plasmids in this repository. Logistics are in progress. With regard to the transgenic flies we are in touch to deposit these 105 fly stocks flies at the Bloomington *Drosophila* Stock centre. This is now mentioned in the manuscript text.

Reviewer #2:The authors have added substantial new validation, particularly on the areas of SNP variation, the lesions obtained, and homozygous phenotypes. This now gives the reader greater confidence in the usefulness of their resource – for somatic knockouts one can often find a localized phenotype but not always, perhaps due to perdurance of a stable gene product; and most of their gRNA stocks can be used to generate germline knockouts, and they have established mutant stocks for about half of them.Taken together, I think this collection will be a valuable resource targeting many genes of great interest for the fly signaling and lipid communities. The paper now gives a much better sense of the approach's strengths and limitations, and there is enough information to make many lines of wide use immediately and to develop further mutant stocks for the remainder. To make these lines accessible to the community, I hope that both the gRNA and mutant stocks will be offered promptly to the major stock centers, if the authors haven't done so already.

We are already in touch with the DGRC to deposit the gRNA plasmids in this repository. Logistics are in progress. With regard to the transgenic flies we are in touch to deposit these 105 fly stocks flies at the Bloomington *Drosophila* Stock centre. This is now mentioned in the manuscript text and the paperwork and logistics is underway.

Reviewer #3:1) The authors have revised the manuscript with additional experiments and clarifications. The revised manuscript appears to be significantly improved by extensive molecular analysis of targeted mutations in fly tissues. Phenotype analysis with additional GAL4 drivers also supports the effectiveness of the CRISPR/Cas9 tools.2) Considering that the majority of targeted genes by ey-GAL4 do not show phenotypes, it is still possible that targeting might not have caused sufficient loss of protein expression in many genes. In this regard, checking the expression level for only one gene (Ocrl) is a weak point. However, the new data showing deletions in most targeted genes seems to compensate for the weakness. In summary, the revised manuscript seems to be acceptable, but a few mistakes listed below should be corrected.3) Some genotypes in figures are not italicized. Those genotypes need to be fixed.

This has been fixed.

4) Anti-Ocrl staining appears to be nuclear staining to me, although it is too weak to be certain. A better way to show the mutation effect is to use a different Gal4 such as en-Gal4 to show an internal control. It would be good to show a test for antibody specificity.

This point is addressed in detail in response to reviewer #1.